# Policy Newton Algorithm in Reproducing Kernel Hilbert Space

**Yixian Zhang**[1]**, Huaze Tang**[1]**, Changxu Wei**[1]**, Chao Wang**[1]**, Wenbo Ding**[1][†]

[1]Tsinghua University

[†]Corresponding Author    { ding.wenbo@sz.tsinghua.edu.cn }

## Abstract

Reinforcement learning (RL) policies represented in Reproducing Kernel Hilbert Spaces (RKHS) offer powerful representational capabilities. While second-order optimization methods like Newton's method demonstrate faster convergence than first-order approaches, current RKHS-based policy optimization remains constrained to first-order techniques. This limitation stems primarily from the intractability of explicitly computing and inverting the infinite-dimensional Hessian operator in RKHS. We introduce Policy Newton in RKHS, the first second-order optimization framework specifically designed for RL policies represented in RKHS. Our approach circumvents direct computation of the inverse Hessian operator by optimizing a cubic regularized auxiliary objective function. Crucially, we leverage the Representer Theorem to transform this infinite-dimensional optimization into an equivalent, computationally tractable finite-dimensional problem whose dimensionality scales with the trajectory data volume. We establish theoretical guarantees proving convergence to a local optimum with a local quadratic convergence rate. Empirical evaluations on a toy financial asset allocation problem validate these theoretical properties, while experiments on standard RL benchmarks demonstrate that Policy Newton in RKHS achieves superior convergence speed and higher episodic rewards compared to established first-order RKHS approaches and parametric second-order methods. Our work bridges a critical gap between non-parametric policy representations and second-order optimization methods in reinforcement learning.

## 1 Introduction

Representing policies within Reproducing Kernel Hilbert Spaces (RKHS) offers a powerful non-parametric alternative, leveraging its definition in an infinite-dimensional functional space to provide strong representational capability and universal approximation (Barreto et al., 2016; Lee et al., 2023). Crucially, the RKHS framework facilitates dynamic complexity adaptation: the policy updates are efficiently restricted to the finite-dimensional span of observed data points, allowing the model size to adapt precisely to the task complexity. These properties are particularly advantageous in data-constrained environments, where sample efficiency is paramount, and in safety-critical applications, where the norm-induced smoothness provides policies with superior robustness and stability against noise and uncertainty (Paternain et al., 2020; Morimura et al., 2010). This approach has demonstrated success in various RL domains, including meta-RL (Lee et al., 2023) and distributional RL (Morimura et al., 2010). Despite these representational advantages, optimization methods for RKHS policies have remained primarily limited to first-order approaches. The RKHS Policy Gradient (Paternain et al., 2020), which achieves policy updates by adding gradient-derived functions in RKHS, represents the current standard. However, this approach inherits the fundamental convergence limitations common to all first-order methods - namely slow convergence in complex optimization landscapes characterized by high curvature or narrow valleys.

In parametric policy representations, second-order optimization methods have emerged as effective solutions to these convergence challenges. While first-order methods like Policy Gradient (Sutton et al., 1999) are widely implemented due to their simplicity, they often exhibit slow convergence and sensitivity to the optimization landscape's curvature, particularly when dealing with ill-conditioned

problems (Furmston et al., 2016). Second-order methods, exemplified by the Policy Newton algorithm (Li et al., 2023; Jha et al., 2020), address these limitations by incorporating Hessian curvature information, enabling potentially faster convergence rates and more appropriately scaled updates. These advantages make second-order methods particularly compelling candidates for accelerating learning in RKHS policy optimization.

The natural progression towards faster optimization—developing a Policy Newton method directly within the RKHS—poses significant theoretical and practical challenges. Unlike the finite-dimensional case, the Hessian analogue in RKHS corresponds to the second-order Fréchet derivative of the expected cumulative reward function. This derivative is an operator acting on the function space, and computing its inverse explicitly, as required by standard Newton methods, is generally intractable in this infinite-dimensional setting. Existing research on second-order methods in RKHS has primarily focused on regret bounds in online learning settings with specific distributed data (Calandriello et al., 2017a;b; Lu et al., 2016; Le et al., 2013), leaving a critical gap for policy optimization in RL where the data distribution shifts with the policy.

To bridge this gap, this paper introduces the **Policy Newton in RKHS** algorithm, the first second-order optimization framework specifically tailored for policies represented within RKHS in the RL context. Our approach circumvents the explicit computation of the infinite-dimensional Hessian operator in policy optimization by reformulating the problem through a cubic regularized auxiliary objective function within the RKHS (Maniyar et al., 2024; Doikov et al., 2024). Crucially, we leverage the Representer Theorem (Schölkopf et al., 2001) to demonstrate that this infinite-dimensional optimization problem is equivalent to solving a finite-dimensional optimization problem in Euclidean space, whose dimension scales with the amount of trajectory data used in the estimate. This makes the approach computationally feasible.

Our main contributions are summarized as follows:

- We propose the first second-order optimization algorithm for policy in RKHS, comprised of two key components: (1) We derive the second-order Fréchet derivative as the Hessian operator and introduce a cubic regularized auxiliary function to find the update step, avoiding the need to compute the intractable inverse operator; (2) We reformulate the infinite-dimensional optimization problem into an equivalent finite-dimensional problem in Euclidean space using the Representer Theorem, making the approach computationally tractable.

- We establish theoretical guarantees for the proposed algorithm, proving convergence to a local optimum, and demonstrating a quadratic convergence rate. Our empirical evaluations on a toy problem verify these theoretical properties and show that Policy Newton in RKHS achieves superior performance in terms of episodic reward compared to baseline methods, with an enhanced ability to escape local optima.

## 2 PRELIMINARIES

### 2.1 POLICY NEWTON IN REINFORCEMENT LEARNING

In reinforcement learning, a Markov decision process is defined by the tuple $(\mathcal{S}, \mathcal{A}, P, r, \gamma, \rho)$ where $\mathcal{S}$ denotes the state space; $\mathcal{A}$ denotes the action space; $P(s_{t+1} \mid s_t, a_t)$ represents the transition probability function; $r(s_t, a_t)$ is the reward function; $\gamma \in [0, 1)$ is the discount factor; and $\rho(s_0)$ is the initial state distribution. Actions are selected according to a policy $\pi(a_t \mid s_t)$, which defines a probability distribution over actions conditional on the current state. A trajectory is denoted by $\omega = (s_0, a_0, \ldots, a_{T-1}, s_T)$, where $s_0 \sim \rho(s_0)$ and $T$ is the episode length. We denote the probability of trajectory $\omega$ following a policy $\pi$ as $p(\omega; \pi)$. Standard reinforcement learning aims to maximize the expected cumulative reward. However, to align with the standard conventions of gradient descent and Newton-type optimization frameworks, we reformulate this as a minimization problem. Throughout this paper, we define the instantaneous cost as the negative reward, effectively setting $r(s_t, a_t) \leftarrow -r(s_t, a_t)$, and minimize the expected discounted cumulative term given by $J(\pi) = \mathbb{E}_{\omega \sim p(\omega; \pi)} \left[ \sum_{t=0}^{T-1} \gamma^{t-1} r(s_t, a_t) \right]$, where $\gamma$ is the discount factor. Typically, the policy is parameterized by a vector $\theta \in \mathbb{R}^d$ and the notation $\pi_\theta$ is used as a shorthand for the distribution

$\pi\left(a_t \mid s_t; \theta\right)$. The target of RL is to find a parameter (Sutton & Barto, 2018)

$$\theta^* = \operatorname*{argmin}_{\theta \in \mathbb{R}^d} J(\pi_\theta).$$

The Policy Gradient method (Maniyar et al., 2024; Williams, 1992) is utilized to find the optimal parameter using the gradient $\nabla_\theta J(\pi_\theta)$ of the expected reward $J(\pi_\theta)$:

$$\nabla_\theta J(\pi_\theta) = \mathbb{E}_{\omega \sim p(\omega;\pi)}\left[\sum_{t=0}^{T-1} \Psi_t(\omega)\nabla_\theta \log \pi\left(a_t \mid s_t; \theta\right)\right],$$

where $\Psi_t(\omega) = \sum_{i=t}^{T-1} \gamma^{i-1} r\left(s_i, a_i\right)$ denotes the cumulative reward starting from $(s_t, a_t)$ in trajectory $\omega$. The Policy Gradient has enjoyed success in many fields, but it is not scale invariant and the search direction is often poorly-scaled (Furmston et al., 2016). To accelerate the optimization, the second-order information is integrated in the Policy Newton method by using the Hessian $\nabla_\theta^2 J(\pi_\theta)$ (Shen et al., 2019):

$$
\begin{aligned}
\nabla_\theta^2 J(\pi_\theta) =& \mathbb{E}_{\omega \sim p(\omega;\pi_\theta)}\left[\sum_{t=0}^{T-1} \Psi_t(\omega)\nabla_\theta \log \pi\left(a_t \mid s_t; \theta\right) \times \sum_{t'=0}^{T-1} \nabla_\theta^\top \log \pi\left(a_{t'} \mid s_{t'}; \theta\right)\right.\\
&\left.+ \sum_{t=0}^{T-1} \Psi_t(\omega)\nabla_\theta^2 \log \pi\left(a_t \mid s_t; \theta\right)\right] = \mathbb{E}_{\omega \sim p(\omega;\pi_\theta)}\left[H_\theta(\omega;\pi_\theta)\right],
\end{aligned}
\tag{1}
$$

where $H_\theta(\omega;\pi)$ represents the Hessian matrix. During the training of the RL, the direct calculation of the gradient is infeasible. Therefore, in the $k$-th iteration of the training, the gradient $\nabla_\theta J(\theta_k)$ is estimated among the sampled trajectory set $\tau$ (Sutton et al., 1999):

$$\nabla_\theta \hat{J}(\theta_k) = \frac{1}{N} \sum_{\omega \in \tau_N} \sum_{t=0}^{T-1} \Psi_t(\omega)\nabla_\theta \log \pi\left(a_t \mid s_t; \theta_k\right), \tag{2}$$

where $N$ denote the size of the trajectory set $\tau_N$. Then the parameter $\theta$ is updated through $\theta_{k+1} = \theta_k + \eta\nabla_\theta \hat{J}(\pi_{\theta_k})$ where $\eta$ is the learning rate. For the Policy Newton method, the Hessian is similarly estimated as $\nabla_\theta^2 \hat{J}(\theta_k) = \frac{1}{N} \sum_{\omega \in \tau_N} H_\theta(\omega;\pi_\theta)$, and the policy is updated through $\theta_{k+1} = \theta_k + \eta[\nabla_\theta^2 \hat{J}(\theta_k)]^{-1}\nabla_\theta \hat{J}(\theta_k)$, where $[\nabla_\theta^2 \hat{J}(\theta_k)]^{-1}\nabla_\theta \hat{J}(\theta_k)$ is the Newton step. The calculation for the inverse of the Hessian is computationally unstable and costly. A direct way to alleviate this drawback is to introduce the regularization term and optimize an auxiliary function to obtain the Newton step (Maniyar et al., 2024; Doikov et al., 2024):

$$\theta_{k+1} = \operatorname*{argmin}_{\theta \in \mathbb{R}^d}\left\{\left\langle\nabla_\theta \hat{J}\left(\theta_k\right), \theta - \theta_k\right\rangle + \frac{1}{2}\left\langle\nabla_\theta^2 \hat{J}\left(\theta_k\right)\left(\theta - \theta_k\right), \theta - \theta_k\right\rangle + \frac{\beta}{6}\left\|\theta - \theta_k\right\|^3\right\}, \tag{3}$$

where $\beta$ is the hyperparameter of the regularization term.

## 2.2 Policy Gradient in RKHS

Reproducing Kernel Hilbert Space (RKHS) is the Hilbert Space $\mathcal{H}_K$ where the element $K(x, \cdot) \in \mathcal{H}_K$ and $f \in \mathcal{H}_K$ satisfy the reproducing property $\langle K(x, \cdot), f\rangle = f(x)$. Despite the policy is modeled by the parameter $\theta$ with particular parameterized functions, the stochastic policy $\pi$ is directly modeled with a function $h$ in RKHS $\mathcal{H}_K$, where the updating gradient for it is also a function (Lever & Stafford, 2015). Particularly, we denote the policy as $\pi_h\left(a_t \mid s_t\right) = \frac{1}{Z} e^{\mathcal{T} h(s_t, a_t)}$ for discrete action space, where $Z = \sum_{a' \in \mathcal{A}} e^{\mathcal{T} h(s_t, a')}$ is the normalization constant and $\mathcal{T}$ is the temperature. Through the definition of the Fréchet derivative (Mcgillivray & Oldenburg, 1990), the Policy Gradient in RKHS is derived as (Mercier et al., 2025; Lever & Stafford, 2015; Paternain et al., 2020):

$$
\begin{aligned}
\nabla_h J(\pi_h) =& \mathbb{E}_{\omega \sim p(\omega;\pi_h)}\left[\mathcal{Z}_h(\omega;\pi_h)\right] = \mathbb{E}_{\omega \sim p(\omega;\pi_h)}\left[\sum_{t=0}^{T-1} \Psi_t(\omega)\nabla_h \log \pi_h\left(a_t \mid s_t\right)\right]\\
=& \mathbb{E}_{\omega \sim p(\omega;\pi_h)}\left[\sum_{t=0}^{T-1} \Psi_t(\omega)\mathcal{T}\left(K\left((s_t, a_t), \cdot\right) - \mathbb{E}_{a' \sim \pi_h(\cdot|s_t)}\left[K\left((s_t, a'), \cdot\right)\right]\right)\right],
\end{aligned}
\tag{4}
$$

where $K\left((s_t, a_t), \cdot\right)$ is the kernel section induced by the state action pair $(s_t, a_t)$. Without loss of generality, the action space is set discretely in the rest of the paper. The estimation of the gradient is

similar to Equation 2, where we denote it as $\nabla_h \hat{J}(\pi_h) = \frac{1}{N} \sum_{\omega \in \tau_N} \mathcal{Z}_h(\omega; \pi)$. Then the policy is updated iteratively by $h_{k+1} = h_k + \eta \nabla_h \hat{J}(\pi_h)$. For simplicity, we denote $\hat{J}(\pi_h)$ as $\hat{J}(h)$ in the rest of this paper.

The use of RKHS policy improves the sample efficiency greatly during training (Paternain et al., 2022), and the overall performance is better than traditional gradient methods (Zhang et al., 2025). However, there is still a lack of research on the Policy Newton algorithm within RKHS. A potential obstacle for its derivation is that the Hessian of $J(\pi_h)$ is infinite, where its inverse is infeasible to represent explicitly. In previous research, Newton optimization is mainly studied in the online learning problem. In (Calandriello et al., 2017a), the Newton optimization scheme is derived where the inverse of the Hessian in RKHS is approximated iteratively. The authors in (Calandriello et al., 2017b) integrate the adaptive embedding with the inverse approximation, which alleviates the computational burden. Despite this success in RKHS Newton optimization, the learning scheme is only suitable for data sampled from the same distribution during training (Lu et al., 2016; Le et al., 2013), while the distribution of transitions in RL is related to the updating policy. As far as we investigated, this is the first paper to derive the Policy Newton in RKHS, and the convergence of our algorithm is also guaranteed.

## 3 THE POLICY NEWTON IN RKHS

In this section, we present the derivation of the Policy Newton in RKHS. In the conventional Policy Newton method, it is simple to obtain the Hessian by deriving the second-order derivative of the expected discounted cumulative reward $J(\pi_\theta)$ with respect to the parameter $\theta$. However, in RKHS space, the second-order Fréchet derivative may not be implicitly represented. Before the derivative of the Hessian within RKHS, we first introduce the following definition:

**Definition 3.1** *Defining the outer product $\mathcal{H}_K \otimes \mathcal{H}_K$ as a new RKHS with operator $\mathbb{K}((s_t, a_t), (s_t', a_t')) = K((s_t, a_t), \cdot) \otimes K((s_t', a_t'), \cdot) \in \mathcal{H}_K \otimes \mathcal{H}_K$ (Kubrusly & Vieira, 2008; Szabó & Sriperumbudur, 2018; Kumari et al., 2017), it satisfies that:*

$$\mathbb{K}((s_t, a_t), (s_t', a_t')) \circ K((s_t'', a_t''), \cdot) = K((s_t, a_t), \cdot) K((s_t', a_t'), (s_t'', a_t''))$$

$$< \mathbb{K}((s_t, a_t), (s_t', a_t')), \mathbb{K}((s_t'', a_t''), (s_t''', a_t''')) > = K((s_t, a_t), (s_t'', a_t'')) K((s_t', a_t'), (s_t''', a_t'''))$$

While the first-order Fréchet derivative $\nabla_h J(h)$ is an element in $\mathcal{H}_K$, the second-order Fréchet derivative is an operator on this space, which we denote by the symbol $\nabla_h^2 J(h)$. This operator can be identified with an element in the tensor product space $\mathcal{H}_K \otimes \mathcal{H}_K$, as shown in the following lemma.

**Lemma 3.1** *The second-order Fréchet derivative $\nabla_h^2 J(\pi_h) = \mathbb{E}_{\omega \sim p(\omega; \pi_h)} [H_h(\omega; \pi_h)]$, where $H_h(\omega; \pi_h)$ is*

$$\left( \sum_{t=0}^{T-1} \Psi_t(\omega) \nabla_h \log \pi_h^t \right) \otimes \left( \sum_{t'=0}^{T-1} \nabla_h^\top \log \pi_h^t \right) - \sum_{t=0}^{T-1} \Psi_t(\omega) \mathcal{T} \operatorname{Cov}_{a' \sim \pi(\cdot|s_t)} [K((s_t, a_t'), \cdot)] .$$

Here $\nabla_h \log \pi_h^t = \nabla_h \log \pi_h(a_t|s_t)$ and $\operatorname{Cov}_{a' \sim \pi(\cdot|s_t)} [K((s_t, a_t'), \cdot)]$ denotes the covariance operator for kernel section $K((s_t, a_t'), \cdot)$, which is detailed as:

$$\mathbb{E}_{a' \sim \pi(\cdot|s_t)} [K((s_t, a'), \cdot) \otimes K((s_t, a'), \cdot)] - \mathbb{E}_{a' \sim \pi(\cdot|s_t)} K((s_t, a'), \cdot) \otimes \mathbb{E}_{a'' \sim \pi(\cdot|s_t)} K((s_t, a''), \cdot) .$$

The detailed derivation is shown in Appendix A. Here, we only introduce a simple example, $U(h) = e^{\mathcal{T} h(s_t, a_t)} K((s_t, a_t), \cdot)$, which is a component in $\nabla_h J(\pi_h)$, to present the core concept for introducing the outer product in RKHS when implementing the Fréchet derivative.

**The second-order Fréchet derivative** Let $h, g \in \mathcal{H}_K$ and $\mathbb{D}(h) = \mathcal{T} e^{\mathcal{T} h(s_t, a_t)} K((s_t, a_t), \cdot) \otimes K((s_t, a_t), \cdot)$. Then according to the definition of the Fréchet derivative (Mcgillivray & Oldenburg, 1990), we testify that $D(h) = \nabla_h U(h)$:

$$\frac{\|U(h+g) - U(h) - \mathbb{D}(h) \circ g\|}{\|g\|} = \frac{\|e^{\mathcal{T} h(x)} K(x, \cdot) [e^{\mathcal{T} g(x)} - 1 - \mathcal{T} g(x)]\|}{\|g\|}$$

$$\leq \frac{e^{\mathcal{T} h(x)} \|K(x, \cdot)\| \frac{M \mathcal{T}^2}{2} |g(x)|^2}{\|g\|} \leq \frac{M \mathcal{T}^2}{2} e^{\mathcal{T} h(x)} (K(x, x))^{3/2} \|g\| \xrightarrow{g \to 0} 0,$$

where the last inequality is due to Cauchy-Schwarz. Through Lemma 3.1, we could find that the second-order Fréchet derivative $\nabla_h^2 J(\pi_h)$ is infeasible to present explicitly. Calculating its inverse is further infeasible for the Policy Newton methods in RKHS. Fundamentally, this is because the Hessian operator is trace-class in the RKHS, and therefore compact. In an infinite-dimensional space, a compact operator does not have a bounded inverse (Kreyszig, 1991), rendering the standard Newton step ill-posed. However, we can still obtain the RKHS Newton step $\Delta h$ through optimizing the regularized auxiliary function similar to Equation 3:

$$\Delta h = \underset{\bar{h} \in \mathcal{H}_K}{\operatorname{argmin}} \left\{ \left\langle \nabla_h \hat{J}(h_k), \bar{h} \right\rangle + \frac{1}{2} \left\langle \nabla_h^2 \hat{J}(h_k) \circ \bar{h}, \bar{h} \right\rangle + \frac{\beta}{6} \left\| \bar{h} \right\|^3 \right\}, \tag{5}$$

where $\nabla_h^2 \hat{J}(h_k) = \frac{1}{N} \sum_{\omega \in \tau_N} H_h(\omega; \pi_h)$ is the estimated RKHS Hessian. Although this estimation is not computationally feasible, the corresponding second-order component $\left\langle \nabla_h^2 \hat{J}(h_k) \circ \bar{h}, \bar{h} \right\rangle$ is easy to calculate according to the Definition 3.1. The optimization in $\mathcal{H}_K$ is still hard to proceed, but through the Representer Theorem in RKHS, we could easily transform the parameter space in this optimization problem from $\mathcal{H}_K$ into $\mathbb{R}$.

**Lemma 3.2** *Representer Theorem (Schölkopf et al., 2001). Suppose we are given a nonempty set $\mathcal{X}$, a positive definite real-valued kernel $K(\cdot, \cdot)$ on $\mathcal{X} \times \mathcal{X}$, a training sample $(x_1, y_1), \ldots, (x_M, y_M) \in \mathcal{X} \times \mathbf{R}$, a strictly monotonically increasing real-valued function $\mathcal{G}$, an arbitrary cost function c. Then any $h \in \mathcal{H}_K$ minimizing the regularized functional*

$$c\left((x_1, y_1, h(x_1)), \ldots, (x_M, y_M, h(x_M))\right) + \mathcal{G}(\|h\|)$$

*admits a representation of the form $h(\cdot) = \sum_{i=1}^{M} \alpha_i K(x_i, \cdot)$, where $\alpha_i$ is the weight for kernel sections.*

Applying the Representer Theorem (Lemma 3.2), Optimization Problem 5 is equivalent to finding $\boldsymbol{\alpha}^*$ via:

$$\boldsymbol{\alpha}^* = \underset{\boldsymbol{\alpha} \in \mathbb{R}^{NT}}{\operatorname{argmin}} \left\{ \left\langle \nabla_h \hat{J}(h_k), \bar{h}_{\boldsymbol{\alpha}} \right\rangle + \frac{1}{2} \left\langle \nabla_h^2 \hat{J}(h_k) \circ \bar{h}_{\boldsymbol{\alpha}}, \bar{h}_{\boldsymbol{\alpha}} \right\rangle + \frac{\beta}{6} \left\| \bar{h}_{\boldsymbol{\alpha}} \right\|^3 \right\}, \tag{6}$$

where $\bar{h}_{\boldsymbol{\alpha}} = \sum_{i=1}^{N} \sum_{t=1}^{T} \alpha_t^i K\left((s_t^i, a_t^i), \cdot\right)$. Here, $(s_t^i, a_t^i)$ denotes the state-action pair for the $i$-th trajectory at time step $t$, and $\boldsymbol{\alpha} = \{\alpha_t^i\}_{i=1, t=1}^{N, T}$ is the set of kernel weights. These weights can be vectorized as $\bar{\boldsymbol{\alpha}} \in \mathbb{R}^{NT}$, where the $l$-th element corresponds to $\alpha_t^i$ with $k = (i-1)T + t$. Applying the Representer Theorem transforms the search for the optimal function perturbation $\bar{h}$ into a search for finite coefficients $\boldsymbol{\alpha}$. By substituting the expansion $\bar{h}_{\boldsymbol{\alpha}}$ back into the operator-based objective Equation 6 and utilizing the reproducing property $\langle K(x, \cdot), K(y, \cdot) \rangle = K(x, y)$, we can explicitly derive the algebraic form of the quadratic and cubic terms with the following theorem.

**Theorem 3.3** *The optimization of the Policy Newton step in RKHS is equal to the optimization of the following quadratic optimization with cubic regularization:*

$$\bar{\boldsymbol{\alpha}}^* = \underset{\bar{\boldsymbol{\alpha}} \in \mathbb{R}^{NT}}{\operatorname{argmin}} \left\{ \langle v, \bar{\boldsymbol{\alpha}} \rangle + \frac{1}{2} \langle H\bar{\boldsymbol{\alpha}}, \bar{\boldsymbol{\alpha}} \rangle + \frac{\beta}{6} \left\| \bar{\boldsymbol{\alpha}} \right\|_2^3 \right\}. \tag{7}$$

*Here, $v \in \mathbb{R}^{NT}$ is the first-order coefficient vector where*

$$v_i = \frac{\mathcal{T}}{N} \sum_{l=1}^{NT} \Psi_l(\omega) \left(K\left((s_l, a_l), (s_i, a_i)\right) - \mathbb{E}_{a'}\left[K\left((s_l, a'), (s_i, a_i)\right)\right]\right).$$

*Let $H \in \mathbb{R}^{NT \times NT}$ be the second-order coefficient matrix given by:*

$$H = \frac{\mathcal{T}^2}{N} bc^\top - \frac{\mathcal{T}}{N} \sum_{l=1}^{NT} \Psi_l(\omega) \Sigma^{(l)}. \tag{8}$$

*Here, $b \in \mathbb{R}^{NT}$ and $c \in \mathbb{R}^{NT}$ are vectors, and $\Sigma^{(l)}$ represents component-related covariance information. The components $b_i$, $c_i$, and the related covariance terms $\Sigma_{ij}^{(l)}$ are defined as:*

$$\begin{cases} b_i = \sum_{l=1}^{NT} \Psi_t(\omega) \left(K_{it} - \mathbb{E}_{a'}[K'_{it}]\right), \qquad c_i = \sum_{l=1}^{NT} \left(K_{il} - \mathbb{E}_{a'}[K'_{il}]\right) \\ \Sigma_{ij}^{(l)} = \operatorname{Cov}_{a' \sim \pi(\cdot | s_l)}\left[K'_{il}, K'_{jl}\right] \end{cases}$$

*In these expressions, the kernel terms are $K_{il} = K\left((s_i, a_i), (s_l, a_l)\right)$ and $K'_{il} = K\left((s_i, a_i), (s_l, a')\right)$.*

The detailed derivation for Theorem 3.3 is presented in Appendix B. It is observed from this theorem that the Policy Newton in RKHS is similar to the traditional Policy Newton method, but the complexity of the Optimization Problem 7 is dependent on the volume of data, i.e., $N \times T$, which possesses the same property of other RKHS methods like support vector machine (Burges, 1998) and radial basis function networks (Park & Sandberg, 1991). We show in the next section the suboptimality and convergence rate for the proposed Policy Newton in RKHS.

**Intuitive Interpretation of the Reduction.**  Before proceeding to the convergence analysis, we briefly clarify the physical meaning of the terms in the finite-dimensional optimization problem 7. The vector $v$ represents the projection of the functional gradient $\nabla_h \hat{J}$ onto the data-dependent subspace spanned by $\{K((s_t^i, a_t^i), \cdot)\}$. The matrix $H$ encapsulates the curvature information: the term $\frac{\mathcal{T}^2}{N} bc^\top$ corresponds to the outer product of first-order gradients (the first term in Equation 8), while the term involving $\Sigma^{(l)}$ captures the covariance structure of the policy's action distribution (the second term).

## 4 THE SUBOPTIMALITY AND CONVERGENCE RATE OF POLICY NEWTON IN RKHS

In this section, we first detail the Policy Newton in RKHS algorithm. We then analyze its convergence properties, demonstrating that despite optimizing via a surrogate function, the resulting policy converges to a local optimum. Furthermore, we show that our proposed Policy Newton in RKHS exhibits a second-order convergence rate, in contrast to the first-order rate achieved by Policy Gradient in RKHS.

### 4.1 THE POLICY NEWTON IN RKHS METHOD

The Optimization Problem 6 admits two primary solution approaches:

(1) Directly computing the derivative of the objective function, setting it to zero, and solving for the critical points.

(2) Optimizing it using various classic optimization methods, including gradient descent, the Newton method, and the conjugate gradient method (Lasdon et al., 2003).

While the analytic method (1) is conceptually simple and direct, it can introduce significant instability into the training process. In complex environments, this instability can lead to exponential error growth. Consequently, method (2) represents a more practical optimization approach. We select the conjugate gradient method as our optimization method. More settings are detailed in Appendix H.

---

**Algorithm 1** Policy Newton RKHS Method

---

**Input:** Number of iterations $M$, trajectory batch size $N$, learning rate $\eta$

1: Initialize RKHS function $h_1 \leftarrow 0$, actor policy $\pi_{h_1}$ based on $h_1$, trajectory set $\tau$.
2: **for** $m = 1, \ldots, M$ **do**
3:     Sample $N$ trajectories using the current policy $\pi_{h_m}$, store in $\tau$.
4:     Estimate the first-order coefficient vector $v$ and second-order coefficient matrix $H$ using $\tau$ (according to Theorem 3.3).
5:     Solve the Optimization Problem 7 using conjugate gradient descent method, output the optimization result $\bar{\alpha}$.
6:     Construct the RKHS update step $\Delta h$ using $\bar{\alpha}$ via Lemma 3.2.
7:     Update the RKHS function: $h_{m+1} \leftarrow h_m + \eta \Delta h$ and the actor policy $\pi_{h_{m+1}}$ based on $h_{m+1}$.
8: **end for**
9: **return** final policy $\pi_{h_{M+1}}$.

---

## 4.2 SUBOPTIMALITY ANALYSIS OF THE PROPOSED ALGORITHM

Theoretically, we assume that the optimal solution to Problem 7 is consistently achieved by Algorithm 1. To establish the convergence properties of our algorithm, we introduce the following lemmas and assumptions.

**Lemma 4.1 (Monte Carlo convergence rate (C.1))** *Assuming* $\mathbb{E}_{\omega \sim p(\omega; \pi_h)}[||\mathcal{Z}_h(\omega; \pi_h)||^2] \leq \sigma_0^2$ *and* $\mathbb{E}_{\omega \sim p(\omega; \pi_h)}[||H_h(\omega; \pi_h)||^2] \leq \sigma_1^2$, *the Monte Carlo estimation of first and second-order Fréchet derivative, namely* $\nabla_h \hat{J}(h_k)$ *and* $\nabla_h^2 \hat{J}(h_k)$, *achieve the convergence rate of* $O(\frac{1}{\sqrt{N}})$:

$$\mathbb{E}_{\omega \sim p(\omega; \pi_h)}\left[||\nabla_h \hat{J}(h_k) - \nabla_h J(h_k)||^2\right] \leq \frac{\sigma_0^2}{N}, \ \mathbb{E}_{\omega \sim p(\omega; \pi_h)}\left[||\nabla_h^2 \hat{J}(h_k) - \nabla_h^2 J(h_k)||^2\right] \leq \frac{\sigma_1^2}{N}.$$

The proof is straightforward by using the property of expectation, which we show in the Appendix C.1. Establishing convergence also requires a Lipschitz continuity assumption for the Hessian, similar to other gradient-based methods (Xiao, 2022; Zhang et al., 2020).

**Assumption 4.1 (Lipschitz continuous)** *The Hessian operator* $\nabla_h^2 J(h)$ *is Lipschitz continuous with constant* $0 \leq L \leq \beta$, *i.e., for all* $h_1, h_2 \in \mathcal{H}_K$: *(we discuss the validation of this assumption in Appendix E)*
$$\|\nabla_h^2 J(h_1) - \nabla_h^2 J(h_2)\| \leq L\|h_1 - h_2\|.$$

Through this assumption, we could establish the upper bound for $J(h)$ with respect to the Hessian and step norm:

**Lemma 4.2 (Taylor upper bound (C.2))** *Under Assumption 4.1, for any* $h_1, h_2 \in \mathcal{H}_K$:

$$J(h_2) \leq J(h_1) + \langle \nabla_h J(h_1), h_2 - h_1 \rangle + \frac{1}{2}\langle \nabla_h^2 J(h_1) \circ (h_2 - h_1), h_2 - h_1 \rangle + \frac{L}{6}\|h_2 - h_1\|^3.$$

The fundamental approach to proving convergence centers on establishing a relationship between the expected gradient norm, $\mathbb{E}\|\nabla_h J(h_k)\|$, and a function denoted by $L(\beta, \sigma_0^2, \sigma_1^2, N)$. Following standard techniques in convergence analysis, the norm of the update step, $\|h_{k+1} - h_k\|$, is employed as an intermediate quantity to construct this relationship. To this end, an upper bound for this step norm is derived in Lemma 4.3.

**Lemma 4.3 (Step norm upper bound (C.3))** *Denoting the updating times for Policy Newton in RKHS as* $M$, *and the number of trajectories sampled in each updating as* $N$, *the updating step can be upper bounded as:*

$$\mathbb{E}\left[\|h_{R+1} - h_R\|^3\right] \leq \frac{36(J(h_1) - J^*)}{\beta M} + \frac{48\sqrt{3}}{\beta^{3/2}}\frac{\sigma_0^{3/2}}{N^{3/4}} + \frac{864}{\beta^3}\frac{\sigma_1^3}{N^{3/2}},$$

*where* $R$ *is a random variable uniformly distributed on* $\{1, \ldots, M\}$, *such that* $P(R = k) = 1/M$.

This lemma provides an upper bound on the expected cubed norm of the update step involving a randomly selected iteration $R$, which is a key quantity used subsequently to establish convergence bounds in expectation for the Policy Newton in RKHS. Next, we establish the lower bound of the norm for the update step, which relates it to the RKHS gradient.

**Lemma 4.4 (Step norm lower bound (C.4))** *The updating step can be lower bounded as:*

$$\mathbb{E}[\|h_{k+1} - h_k\|^2] \geq \frac{1}{L + \beta}\left(\mathbb{E}[\|\nabla J(h_{k+1})\|] - \frac{\sigma_0}{\sqrt{N}} - \frac{\sigma_1^2}{2N(L + \beta)}\right).$$

Having established both lower and upper bounds for the step norm, we can now use these results to construct the main convergence theorem.

**Theorem 4.5 (Convergence property (C.5))** *Given Lemmas 4.3 and 4.4, let* $R$ *be a random variable uniformly distributed on* $\{1, \ldots, M\}$. *The sequence* $\{h_k\}$ *generated by iterative optimization in Theorem 3.3 satisfies*
$$\lim_{M, N \to \infty} \mathbb{E}[\|\nabla J(h_{R+1})\|] = 0.$$

This theorem indicates that the expected gradient norm at a randomly chosen iteration converges to zero, implying convergence towards a stationary point.

### 4.3 The second-order convergence rate

Establishing the convergence rate for stochastic Policy Newton methods typically requires specific assumptions regarding problem structure. While the convergence properties of Newton's method in finite-dimensional Euclidean spaces are well-established (Furmston et al., 2016), extending these guarantees to the infinite-dimensional RKHS setting—specifically with the cubic regularization auxiliary function—requires rigorous verification. We demonstrate that the finite-dimensional reduction derived in Theorem 3.3 preserves the desirable quadratic convergence properties of the original operator-theoretic problem. To establish baseline performance characteristics, this section analyzes the convergence rate under idealized conditions, while a comprehensive analysis under more realistic stochastic assumptions remains for future work. Specifically, we assume access to the true gradient $\nabla_h J(h_k)$ and Hessian $\nabla_h^2 J(h_k)$ at each iteration $k$, effectively setting $\nabla_h \hat{J}(h_k) = \nabla_h J(h_k)$ and $\nabla_h^2 \hat{J}(h_k) = \nabla_h^2 J(h_k)$. Under this deterministic scenario, we prove that the method achieves a local quadratic convergence rate.

Our analysis relies on a key assumption: that the inverse of the regularized Hessian is uniformly bounded. This is a standard condition in the analysis of Newton-type methods, required to ensure the update step is well-defined by excluding potential singularities (Nesterov & Polyak, 2006a; Nocedal & Wright, 2006a).

**Theorem 4.6 (Local Quadratic Convergence (D))** *Consider the deterministic Policy Newton RKHS method (Algorithm 1 with $\nabla_h \hat{J}(h_k) = \nabla_h J(h_k)$ and $\nabla_h^2 \hat{J}(h_k) = \nabla_h^2 J(h_k)$). Assuming the norm of the inverse operator $\|(\nabla_h^2 J(h_k) + \frac{\beta}{2}\|\Delta h_k\|\mathcal{I})^{-1}\|$ is bounded by some constant $B$, and we assume that the update step is sufficiently small that $\|\Delta h_k\| \leq L\|e_k\|$.*

*If the initial iterate $h_0$ is sufficiently close to $h^*$, the sequence $\{h_k\}$ converges quadratically to $h^*$. That is, there exists a constant $C_q > 0$ such that*

$$\|h_{k+1} - h^*\| \leq C_q \|h_k - h^*\|^2$$

*for all $k$ sufficiently large. The validation of the assumptions is discussed in Appendix E.*

## 5 Numerical experiment

This section presents an empirical evaluation of our proposed Policy Newton method in Reproducing Kernel Hilbert Space (RKHS) across two distinct experimental settings: (a) a simplified Asset Allocation environment designed specifically to demonstrate the quadratic convergence properties of Policy Newton in RKHS (Yoo et al., 2023), and (b) complex control tasks from the Gymnasium framework (Towers et al., 2024), including CartPole and Lunar Lander. Throughout all experiments, we utilize a standard Gaussian kernel for the RKHS representations. The Asset Allocation environment serves to empirically validate the theoretical convergence guarantees established in Section 4. Additionally, we benchmark Policy Newton in RKHS against several baseline methods in complex environments to demonstrate its superior performance characteristics.

### 5.1 Quadratic convergence tested in the toy experiment

We empirically validate the quadratic convergence properties of Policy Newton in RKHS using a simplified Asset Allocation environment (Lee et al., 2004; Yoo et al., 2023). While the complete asset allocation problem presents substantial analytical challenges, we utilize a reduced-complexity variant (detailed in Appendix F) where the global optimal policy can be explicitly represented, enabling precise quantification of convergence properties for both the policy and cumulative reward $J(\pi)$.

For comparative analysis, we implemented four distinct methodologies. The conventional Policy Gradient and Policy Newton methods (Maniyar et al., 2024) utilize discrete policies with parameterized action probabilities. The Policy Gradient in RKHS implementation follows the approach described in (Lever & Stafford, 2015), while our Policy Newton in RKHS method is implemented according to Algorithm 1. All policies were initialized with uniform distributions.

The experimental results presented in Figure 1 reveal several important findings. Figure 1a demonstrates that both Policy Gradient in RKHS and Policy Newton in RKHS converge rapidly toward the maximum expected episodic reward. Notably, Policy Newton in RKHS exhibits clear quadratic

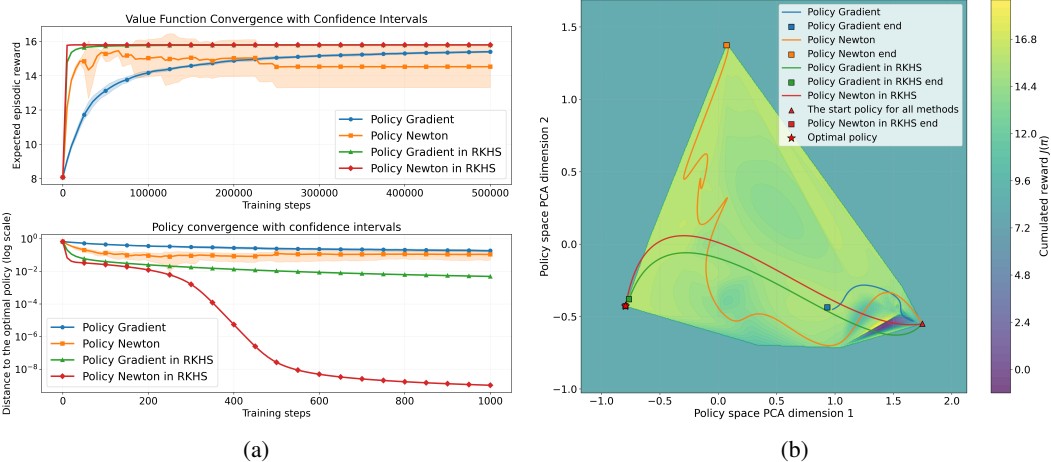

Figure 1: Experimental results demonstrating the quadratic convergence of Policy Newton in RKHS within the simplified Asset Allocation environment. Three benchmark methods are compared: conventional Policy Gradient, Policy Newton, and Policy Gradient in RKHS. 1a illustrates the convergence metrics during training. 1b visualizes the optimization trajectories of all methods in a PCA-projected policy space, where the background surface is an interpolated return landscape. This plot highlights how different algorithms move through the policy space toward the optimal policy.

convergence behavior as it approaches the optimal policy. In contrast, the conventional methods show different characteristics—while the standard Policy Newton method converges more rapidly than conventional Policy Gradient (as shown in Figures 1a and 1b), its training trajectory exhibits greater instability and ultimately converges to a suboptimal local maximum. It is important to note that in this toy environment all four compared methods share the same representational capacity: the state and action spaces are finite, and each algorithm directly optimizes the probability value assigned to every (state, action) pair. Thus, all methods are able to exactly represent the optimal policy. The differences observed in Figure 1b therefore arise purely from the optimization geometry rather than the expressiveness of the policy class. In particular, RKHS-based updates span the data-dependent kernel basis, which yields a richer set of descent directions and enables the optimizer to move out of suboptimal attraction regions that trap parametric methods.

## 5.2 TRAINING PERFORMANCE IN RL TESTING ENVIRONMENT

To evaluate the efficacy and universality of the proposed Policy Newton in RKHS algorithm, we conduct experiments utilizing standard RL environments from the Gymnasium suite (Towers et al., 2024), covering both discrete and continuous action spaces.

**Discrete Control Tasks.** We first evaluate the method on CartPole and Lunar Lander. These environments feature discrete action spaces, directly utilizing the theoretical framework established in Section 3. The baseline policies (Policy Gradient and Policy Newton) are parameterized using a linear model augmented with a polynomial transformation (Maniyar et al., 2024) to ensure fair comparison in representational power.

**Continuous Control Tasks.** To demonstrate the method's scalability to high-dimensional continuous control, we further evaluate it on Inverted Pendulum and Hopper. For these tasks, we implement the Policy Newton in RKHS using the continuous Gaussian policy formulation. The rigorous theoretical derivation for the second-order RKHS step in the Gaussian policy is provided in Appendix I. The baselines utilize the same polynomial feature expansion as in the discrete case but map to continuous action outputs.

**Performance Analysis.** The experimental results are summarized in Figure 2. In the discrete tasks (Figs. 2a and 2b), Policy Newton in RKHS exhibits rapid convergence and superior sample efficiency, significantly outperforming the first-order RKHS baseline and the parametric Policy Newton method.

In the continuous tasks (Figs. 2c and 2d), the advantage of our method is equally pronounced. In Inverted Pendulum, our method stabilizes quickly. In the challenging Hopper environment, Policy Newton in RKHS demonstrates superior sample efficiency, achieving high rewards with fewer iterations than the baselines. We attribute this performance to the effective utilization of curvature information via the Hessian operator, which aids in navigating complex optimization landscapes, and the flexible representational capacity of the RKHS.

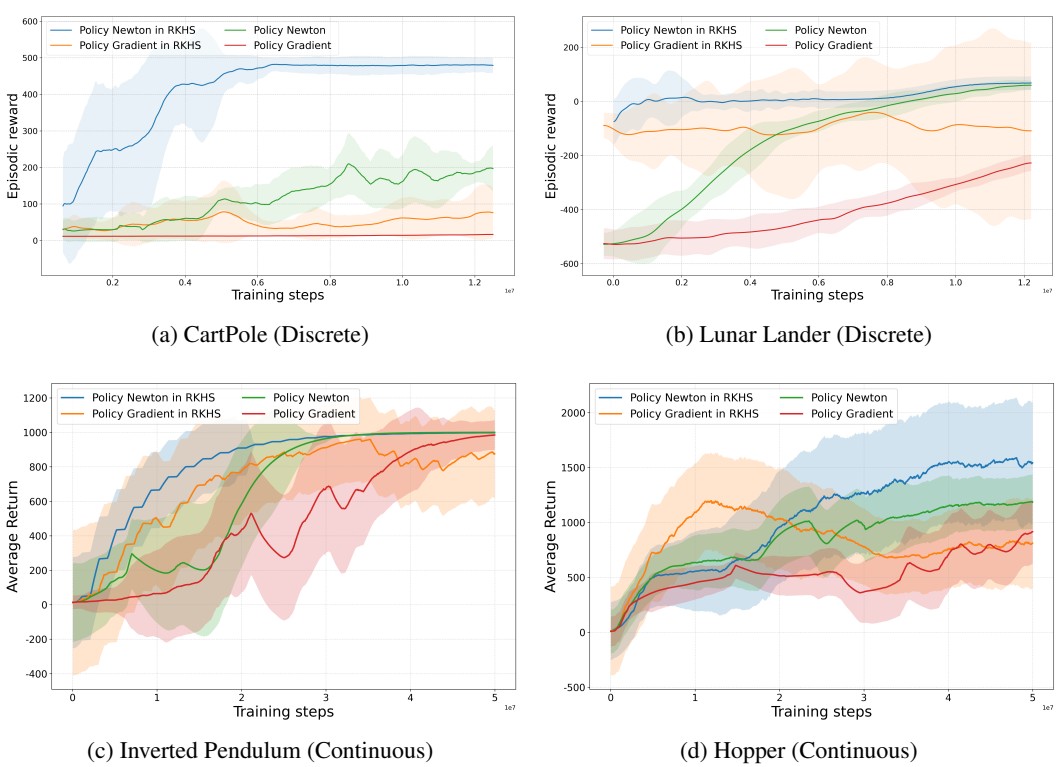

(a) CartPole (Discrete)

(b) Lunar Lander (Discrete)

(c) Inverted Pendulum (Continuous)

(d) Hopper (Continuous)

Figure 2: Comparative analysis of Policy Newton in RKHS against established baseline methods across discrete and continuous Gymnasium environments. The plots display mean episodic rewards across 5 independent runs, and the shaded regions denote the 95% confidence intervals. (a-b) Performance on discrete tasks, utilizing the formulation in Section 3. (c-d) Performance on continuous control tasks, utilizing the extension derived in Appendix I. Our method demonstrates consistent superior sample efficiency and rapid convergence speed in both settings.

## 6 CONCLUSION AND FUTURE WORK

This paper successfully introduced Policy Newton in RKHS, the first practical second-order optimization method for reinforcement learning policies represented within Reproducing Kernel Hilbert Spaces. We established its theoretical foundations, proving convergence to a local optimum and demonstrating a local quadratic convergence rate. These theoretical properties were empirically validated on both a toy problem and standard RL benchmarks, where Policy Newton in RKHS achieved significantly faster convergence to superior episodic rewards compared to first-order and parametric Newton baselines.

While the current results are promising, extending the application of Policy Newton in RKHS to highly complex RL problems, such as the Humanoid environment (Todorov et al., 2012), may reveal challenges related to robustness and stability. A promising avenue for future research is the integration of neural networks with the Policy Newton in RKHS framework, potentially drawing inspiration from architectures similar to those proposed in (Zhang et al., 2025), to enhance performance in such demanding scenarios. The primary focus of this paper has been the rigorous theoretical establishment of Policy Newton in RKHS, laying the groundwork for these and other exciting explorations in future work.

## ETHICS STATEMENT

This study uses only a synthetic toy Asset Allocation environment (Appendix F) and public Gymnasium benchmarks (CartPole, Lunar Lander) (Towers et al., 2024); no human subjects, personal data, or sensitive attributes are involved, and no proprietary datasets are introduced. We caution against deploying the method in high-stakes settings (e.g., financial decision systems) without domain-specific governance, distribution-shift monitoring, and independent validation. The compute footprint is modest. The authors disclose no conflicts of interest and no external sponsorship that could bias the work.

## REPRODUCIBILITY STATEMENT

We provide an anonymous supplementary archive containing the full implementation, dependency specifications, and runnable scripts to reproduce all figures. The algorithmic procedure is given in Algorithm 1; the finite-dimensional reduction and coefficient construction appear in Theorem 3.3, with derivations in Appendix B and RKHS second-order calculus in Appendix A. Experimental settings, hyperparameters, and random seeds are documented in Appendix H; the toy environment is fully specified in Appendix F. These materials enable end-to-end reproduction of the reported results.

### ACKNOWLEDGMENTS

This work was supported by Shenzhen Science and Technology Program (No. KJZD20240903100905008) and Ant Group.

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

## A  THE DERIVATION OF THE SECOND-ORDER FRÉCHET DERIVATIVE IN RKHS

This section details the derivation of second-order Fréchet derivatives of the log-policy, $\log \pi_h(a_t \mid s_t)$, with respect to the function $h \in \mathcal{H}_K$. The first-order Fréchet derivative is introduced in (Mercier et al., 2025), for convenience's sake, we also detail it in this section. The policy is defined as $\pi_h(a_t \mid s_t) = \frac{1}{Z} e^{\mathcal{T} h(s_t, a_t)}$, where $Z = \sum_{a' \in \mathcal{A}} e^{\mathcal{T} h(s_t, a')}$ is the normalization constant. For brevity in this section, we will denote $(s_t, a_t)$ as $(s, a)$ when the context is clear for a single state-action pair for which the log-policy is being differentiated. The kernel section $K((s, a), \cdot)$ is an element in $\mathcal{H}_K$.

We first derive the first-order Fréchet derivative $\nabla_h \log \pi_h(a \mid s)$. The log-policy is $\log \pi_h(a \mid s) = \mathcal{T} h(s, a) - \log Z$.

The Fréchet derivative of the first term is:

$$\nabla_h(\mathcal{T} h(s, a)) = \mathcal{T} K((s, a), \cdot)$$

For the second term, $-\log Z$:

$$\nabla_h(-\log Z) = -\frac{1}{Z} \nabla_h Z$$

We compute $\nabla_h Z$:

$$Z = \sum_{a' \in \mathcal{A}} e^{\mathcal{T} h(s, a')}$$

$$\nabla_h Z = \sum_{a'} \nabla_h(e^{\mathcal{T}h(s,a')}) = \sum_{a'} e^{\mathcal{T}h(s,a')} \mathcal{T} \nabla_h h(s,a') = \mathcal{T} \sum_{a'} e^{\mathcal{T}h(s,a')} K((s,a'),\cdot)$$

Substituting this back:

$$\nabla_h(-\log Z) = -\frac{1}{Z} \mathcal{T} \sum_{a'} e^{\mathcal{T}h(s,a')} K((s,a'),\cdot)$$

Recognizing that $\frac{e^{\mathcal{T}h(s,a')}}{Z} = \pi_h(a' \mid s)$, we simplify:

$$\nabla_h(-\log Z) = -\mathcal{T} \sum_{a'} \pi_h(a' \mid s) K((s,a'),\cdot) = -\mathcal{T} \mathbb{E}_{a' \sim \pi_h(\cdot|s)} [K((s,a'),\cdot)]$$

Combining the derivatives of both terms, we obtain the first-order Fréchet derivative of the log-policy:

$$\nabla_h \log \pi_h(a \mid s) = \mathcal{T} K((s,a),\cdot) - \mathcal{T} \mathbb{E}_{a' \sim \pi_h(\cdot|s)} [K((s,a'),\cdot)]$$

Factorizing $\mathcal{T}$:

$$\nabla_h \log \pi_h(a \mid s) = \mathcal{T} \left( K((s,a),\cdot) - \mathbb{E}_{a' \sim \pi_h(\cdot|s)} [K((s,a'),\cdot)] \right) \tag{9}$$

This expression forms the core of the Policy Gradient in RKHS as shown in Equation 4 of the main paper when appropriately weighted and summed.

Next, we derive the second-order Fréchet derivative (Hessian operator) of the log-policy with respect to $h$, denoted as $\nabla_h^2 \log \pi_h(a \mid s)$. This is obtained by differentiating Equation 9:

$$\nabla_h^2 \log \pi_h(a \mid s) = \nabla_h \left[ \mathcal{T} \left( K((s,a),\cdot) - \sum_{a' \in \mathcal{A}} \pi_h(a' \mid s) K((s,a'),\cdot) \right) \right]$$

Since $\mathcal{T}$ is a constant and $K((s,a),\cdot)$ is a fixed element in $\mathcal{H}_K$ (not depending on $h$ for this differentiation), its derivative is zero:

$$\nabla_h^2 \log \pi_h(a \mid s) = \mathcal{T} \nabla_h \left( -\sum_{a' \in \mathcal{A}} \pi_h(a' \mid s) K((s,a'),\cdot) \right)$$

Applying the product rule for Fréchet derivatives (treating $K((s,a'),\cdot)$ as a constant vector in $\mathcal{H}_K$ for each $a'$):

$$\nabla_h^2 \log \pi_h(a \mid s) = -\mathcal{T} \sum_{a' \in \mathcal{A}} (\nabla_h \pi_h(a' \mid s)) \otimes K((s,a'),\cdot)$$

Here, $\otimes$ denotes the outer product as defined in Definition 3.1.

Now, we compute $\nabla_h \pi_h(a' \mid s)$. Recall $\pi_h(a' \mid s) = \frac{e^{\mathcal{T}h(s,a')}}{Z}$. Using the quotient rule $\nabla_h(\frac{N}{D}) = \frac{(\nabla_h N)D - N(\nabla_h D)}{D^2}$: Let $N_{a'} = e^{\mathcal{T}h(s,a')}$, so $\nabla_h N_{a'} = \mathcal{T} e^{\mathcal{T}h(s,a')} K((s,a'),\cdot)$. Let $D = Z = \sum_{a''} e^{\mathcal{T}h(s,a'')}$, so $\nabla_h D = \mathcal{T} \sum_{a''} e^{\mathcal{T}h(s,a'')} K((s,a''),\cdot)$.

$$\nabla_h \pi_h(a' \mid s) = \frac{\left( \mathcal{T} e^{\mathcal{T}h(s,a')} K((s,a'),\cdot) \right) Z - e^{\mathcal{T}h(s,a')} \left( \mathcal{T} \sum_{a''} e^{\mathcal{T}h(s,a'')} K((s,a''),\cdot) \right)}{Z^2}$$

$$= \mathcal{T} \frac{e^{\mathcal{T}h(s,a')}}{Z} K((s,a'),\cdot) - \mathcal{T} \frac{e^{\mathcal{T}h(s,a')}}{Z} \frac{\sum_{a''} e^{\mathcal{T}h(s,a'')} K((s,a''),\cdot)}{Z}$$

$$= \mathcal{T} \pi_h(a' \mid s) K((s,a'),\cdot) - \mathcal{T} \pi_h(a' \mid s) \sum_{a''} \pi_h(a'' \mid s) K((s,a''),\cdot)$$

$$\nabla_h \pi_h(a' \mid s) = \mathcal{T} \pi_h(a' \mid s) \left( K((s,a'),\cdot) - \mathbb{E}_{a'' \sim \pi_h(\cdot|s)} [K((s,a''),\cdot)] \right)$$

Substituting this expression for $\nabla_h \pi_h(a' \mid s)$ back into the equation for $\nabla_h^2 \log \pi_h(a \mid s)$. To align with the result in Lemma 3.1 of the main paper, which states $\nabla_h^2 \log \pi_h(a_t \mid s_t) =$

$-\mathcal{T}\operatorname{Cov}_{a'\sim\pi_h(\cdot|s_t)}[K((s_t,a'),\cdot)]$, the substitution effectively uses $\nabla_h\pi_h(a'\mid s)/\mathcal{T}$:

$$
\begin{aligned}
\nabla_h^2\log\pi_h(a\mid s) &= -\mathcal{T}\sum_{a'\in\mathcal{A}}\pi_h(a'\mid s)\left(K((s,a'),\cdot)-\mathbb{E}_{a''\sim\pi_h(\cdot|s)}\left[K((s,a''),\cdot)\right]\right)\otimes K((s,a'),\cdot)\\
&= -\mathcal{T}\left(\sum_{a'\in\mathcal{A}}\pi_h(a'\mid s)K((s,a'),\cdot)\otimes K((s,a'),\cdot)\right.\\
&\quad\left. -\sum_{a'\in\mathcal{A}}\pi_h(a'\mid s)\left(\mathbb{E}_{a''\sim\pi_h(\cdot|s)}\left[K((s,a''),\cdot)\right]\right)\otimes K((s,a'),\cdot)\right)\\
&= -\mathcal{T}\left(\mathbb{E}_{a'\sim\pi_h(\cdot|s)}\left[K((s,a'),\cdot)\otimes K((s,a'),\cdot)\right]\right.\\
&\quad\left. -\left(\mathbb{E}_{a''\sim\pi_h(\cdot|s)}\left[K((s,a''),\cdot)\right]\right)\otimes\left(\sum_{a'\in\mathcal{A}}\pi_h(a'\mid s)K((s,a'),\cdot)\right)\right)\\
&= -\mathcal{T}\left(\mathbb{E}_{a'\sim\pi_h(\cdot|s)}\left[K((s,a'),\cdot)\otimes K((s,a'),\cdot)\right]\right.\\
&\quad\left. -\mathbb{E}_{a'\sim\pi_h(\cdot|s)}\left[K((s,a'),\cdot)\right]\otimes\mathbb{E}_{a''\sim\pi_h(\cdot|s)}\left[K((s,a''),\cdot)\right]\right)
\end{aligned}
\tag{10}
$$

This can be compactly written using the covariance operator as defined in Lemma 3.1 (using $s_t, a_t$ for generality):

$$
\nabla_h^2\log\pi_h(a_t\mid s_t) = -\mathcal{T}\operatorname{Cov}_{a'\sim\pi_h(\cdot|s_t)}\left[K((s_t,a'),\cdot)\right]
$$

This expression for $\nabla_h^2\log\pi_h(a_t\mid s_t)$ is the component used in constructing the Hessian operator in Lemma 3.1 and the estimated Hessian $\nabla_h^2\hat{J}(h_k)$ in the paper. By substituting the first and second derivative in 1 with $\nabla_h\log\pi_h(a_t\mid s_t)$ and $\nabla_h^2\log\pi_h(a_t\mid s_t)$, the Lemma 3.1 is proved.

# B  THE DERIVATION OF THEOREM 3.3

Theorem 3.3 transforms the RKHS optimization problem for the Newton step $\Delta h$ (Equation 5) into an equivalent finite-dimensional optimization problem (Equation 7). The RKHS update step is $\Delta h = \bar{h}_{\boldsymbol{\alpha}}(\cdot) = \sum_{k=1}^{NT}\alpha_k K(x_k,\cdot)$, where $x_k = (s_k, a_k)$ are state-action pairs from the $N\times T$ trajectory data points ($k$ is a flattened index from 1 to $M=NT$), and $\bar{\boldsymbol{\alpha}}\in\mathbb{R}^{NT}$ is the coefficient vector.

The objective function in Equation 5 is:

$$
L(\bar{\boldsymbol{\alpha}}) = \left\langle\nabla_h\hat{J}(h_k),\bar{h}_{\boldsymbol{\alpha}}\right\rangle + \frac{1}{2}\left\langle\nabla_h^2\hat{J}(h_k)\circ\bar{h}_{\boldsymbol{\alpha}},\bar{h}_{\boldsymbol{\alpha}}\right\rangle + \frac{\beta}{6}\|\bar{\boldsymbol{\alpha}}\|_2^3
$$

We derive the forms for the first two terms. The third term, $\frac{\beta}{6}\|\bar{\boldsymbol{\alpha}}\|_2^3$, directly uses the Euclidean norm of $\bar{\boldsymbol{\alpha}}$ as stated in Equation 7.

Let $M = NT$. The set of basis functions is $\{K(x_k,\cdot)\}_{k=1}^M$. The perturbation is $\bar{h}_{\boldsymbol{\alpha}}(\cdot) = \sum_{i=1}^M\alpha_i K(x_i,\cdot)$. We use index $i$ (or $j$) for the coefficients $\alpha_i$ and the basis functions $K(x_i,\cdot)$. We use index $l$ (or $l'$) for data points from the batch of $M$ samples when defining the gradient and Hessian operators.

## B.1  FIRST-ORDER TERM: $\left\langle\nabla_h\hat{J}(h_k),\bar{h}_{\boldsymbol{\alpha}}\right\rangle$

The estimated first-order Fréchet derivative $\nabla_h\hat{J}(h_k)$ (denoted $g_{op}$ for operator form) is given by adapting Equation 4 for the empirical average over $N$ trajectories, or $M=NT$ total samples:

$$
g_{op}(\cdot) = \nabla_h\hat{J}(h_k)(\cdot) = \frac{\mathcal{T}}{N}\sum_{l=1}^M\Psi_l(\omega)\left(K(x_l,\cdot)-\mathbb{E}_{a'\sim\pi(\cdot|s_l)}\left[K((s_l,a'),\cdot)\right]\right)
$$

where $x_l = (s_l, a_l)$ is the $l$-th data point in the batch, and $\Psi_l(\omega)$ is its associated cumulative reward. The inner product is:

$$\left\langle \nabla_h \hat{J}(h_k), \bar{h}_{\boldsymbol{\alpha}} \right\rangle = \left\langle \nabla_h \hat{J}(h_k), \sum_{i=1}^{M} \alpha_i K(x_i, \cdot) \right\rangle$$

$$= \sum_{i=1}^{M} \alpha_i \left\langle \nabla_h \hat{J}(h_k), K(x_i, \cdot) \right\rangle \quad \text{(by linearity of inner product)}$$

Let $v_i = \left\langle \nabla_h \hat{J}(h_k), K(x_i, \cdot) \right\rangle$. Using the reproducing property and the expression for $\nabla_h \hat{J}(h_k)$:

$$v_i = \frac{\mathcal{T}}{N} \sum_{l=1}^{M} \Psi_l(\omega) \left\langle K(x_l, \cdot) - \mathbb{E}_{a' \sim \pi(\cdot|s_l)} \left[ K((s_l, a'), \cdot) \right], K(x_i, \cdot) \right\rangle$$

$$= \frac{\mathcal{T}}{N} \sum_{l=1}^{M} \Psi_l(\omega) \left( K(x_l, x_i) - \mathbb{E}_{a' \sim \pi(\cdot|s_l)} \left[ K((s_l, a'), x_i) \right] \right)$$

Thus, $\left\langle \nabla_h \hat{J}(h_k), \bar{h}_{\boldsymbol{\alpha}} \right\rangle = \sum_{i=1}^{M} \alpha_i v_i = v^\top \bar{\boldsymbol{\alpha}}$. This definition of $v_i$ matches Theorem 3.3.

## B.2 Second-Order Term: $\frac{1}{2} \left\langle \nabla_h^2 \hat{J}(h_k) \circ \bar{h}_{\boldsymbol{\alpha}}, \bar{h}_{\boldsymbol{\alpha}} \right\rangle$

Let $H_{op}^{est} = \nabla_h^2 \hat{J}(h_k)$. Based on the updated Lemma 3.1 and Equation 1, the estimated Hessian operator $\nabla_h^2 \hat{J}(h_k)$ includes two components:

$$\frac{1}{N} \left[ \left( \sum_{l=1}^{M} \Psi_l(\omega) \nabla_h \log \pi_h(x_l) \right) \otimes \left( \sum_{l'=1}^{M} \nabla_h \log \pi_h(x_{l'}) \right) - \sum_{l=1}^{M} \Psi_l(\omega) \mathcal{T} \operatorname{Cov}_{a' \sim \pi(\cdot|s_l)} \left[ K(x_l, a') \right] \right]$$

Let $H_{op}^{(1)}$ be the operator for the first part (outer product) and $H_{op}^{(2)}$ for the second part (covariance sum), such that $\nabla_h^2 \hat{J}(h_k) = \frac{1}{N}(H_{op}^{(1)} - H_{op}^{(2)})$.

**1. Contribution from $H_{op}^{(1)}$:** Let $\nabla_h \log \pi_h(x_l)(\cdot) = \mathcal{T} \left( K(x_l, \cdot) - \mathbb{E}_{a' \sim \pi(\cdot|s_l)} \left[ K((s_l, a'), \cdot) \right] \right)$. Let $X_l(\bar{h}_{\boldsymbol{\alpha}}) = \langle \nabla_h \log \pi_h(x_l), \bar{h}_{\boldsymbol{\alpha}} \rangle$.

$$X_l(\bar{h}_{\boldsymbol{\alpha}}) = \mathcal{T} \sum_{i=1}^{M} \alpha_i \left( K(x_l, x_i) - \mathbb{E}_{a' \sim \pi(\cdot|s_l)} \left[ K((s_l, a'), x_i) \right] \right)$$

The quadratic form from $H_{op}^{(1)}$ is $\langle H_{op}^{(1)} \circ \bar{h}_{\boldsymbol{\alpha}}, \bar{h}_{\boldsymbol{\alpha}} \rangle = \left( \sum_{l=1}^{M} \Psi_l(\omega) X_l(\bar{h}_{\boldsymbol{\alpha}}) \right) \left( \sum_{l'=1}^{M} X_{l'}(\bar{h}_{\boldsymbol{\alpha}}) \right)$. Using the definitions of $b_i$ and $c_i$ from Theorem 3.3 (with $K_{il} = K(x_i, x_l)$ and $K'_{il} = K(x_i, (s_l, a'))$, and summation index $l$ for data points):

$$b_i = \sum_{l=1}^{M} \Psi_l(\omega) \left( K(x_i, x_l) - \mathbb{E}_{a' \sim \pi(\cdot|s_l)} \left[ K(x_i, (s_l, a')) \right] \right)$$

$$c_i = \sum_{l=1}^{M} \left( K(x_i, x_l) - \mathbb{E}_{a' \sim \pi(\cdot|s_l)} \left[ K(x_i, (s_l, a')) \right] \right)$$

Then, $\sum_{l=1}^{M} \Psi_l(\omega) X_l(\bar{h}_{\boldsymbol{\alpha}}) = \mathcal{T} \sum_{i=1}^{M} \alpha_i b_i = \mathcal{T}(\bar{\boldsymbol{\alpha}}^\top b)$. And, $\sum_{l'=1}^{M} X_{l'}(\bar{h}_{\boldsymbol{\alpha}}) = \mathcal{T} \sum_{j=1}^{M} \alpha_j c_j = \mathcal{T}(\bar{\boldsymbol{\alpha}}^\top c)$. So, the contribution to $\left\langle \nabla_h^2 \hat{J}(h_k) \circ \bar{h}_{\boldsymbol{\alpha}}, \bar{h}_{\boldsymbol{\alpha}} \right\rangle$ from this first part is:

$$\frac{1}{N} \langle H_{op}^{(1)} \circ \bar{h}_{\boldsymbol{\alpha}}, \bar{h}_{\boldsymbol{\alpha}} \rangle = \frac{1}{N} (\mathcal{T} \bar{\boldsymbol{\alpha}}^\top b)(\mathcal{T} c^\top \bar{\boldsymbol{\alpha}}) = \bar{\boldsymbol{\alpha}}^\top \left( \frac{\mathcal{T}^2}{N} bc^\top \right) \bar{\boldsymbol{\alpha}}$$

**2. Contribution from $H_{op}^{(2)}$:** $H_{op}^{(2)} \circ u = \sum_{l=1}^{M} \Psi_l(\omega) \mathcal{T} \operatorname{Cov}_{a' \sim \pi(\cdot|s_l)} [K((s_l, a'), \cdot)] \circ u$. The quadratic form is $\langle H_{op}^{(2)} \circ \bar{h}_{\boldsymbol{\alpha}}, \bar{h}_{\boldsymbol{\alpha}} \rangle = \sum_{l=1}^{M} \Psi_l(\omega) \mathcal{T} \left\langle \operatorname{Cov}_{a' \sim \pi(\cdot|s_l)} [K((s_l, a'), \cdot)] \circ \bar{h}_{\boldsymbol{\alpha}}, \bar{h}_{\boldsymbol{\alpha}} \right\rangle$.

The inner term is $\mathrm{Var}_{a'\sim\pi(\cdot|s_l)}[\langle K((s_l,a'),\cdot),\bar{h}_{\boldsymbol{\alpha}}\rangle] = \bar{\boldsymbol{\alpha}}^\top\Sigma^{(l)}\bar{\boldsymbol{\alpha}}$, where $\Sigma^{(l)}_{ij} = \mathrm{Cov}_{a'\sim\pi(\cdot|s_l)}[K((s_l,a'),x_i),K((s_l,a'),x_j)]$. Using $K'_{il} = K(x_i,(s_l,a'))$, this is $\Sigma^{(l)}_{ij} = \mathrm{Cov}_{a'\sim\pi(\cdot|s_l)}\left[K'_{il},K'_{jl}\right]$, matching Theorem 3.3. So, the contribution to $\left\langle\nabla^2_h\hat{J}(h_k)\circ\bar{h}_{\boldsymbol{\alpha}},\bar{h}_{\boldsymbol{\alpha}}\right\rangle$ from this second part is:

$$-\frac{1}{N}\langle H^{(2)}_{op}\circ\bar{h}_{\boldsymbol{\alpha}},\bar{h}_{\boldsymbol{\alpha}}\rangle = -\frac{1}{N}\bar{\boldsymbol{\alpha}}^\top\left(\mathcal{T}\sum_{l=1}^M\Psi_l(\omega)\Sigma^{(l)}\right)\bar{\boldsymbol{\alpha}} = \bar{\boldsymbol{\alpha}}^\top\left(-\frac{\mathcal{T}}{N}\sum_{l=1}^M\Psi_l(\omega)\Sigma^{(l)}\right)\bar{\boldsymbol{\alpha}}$$

**Combining terms for the matrix $H$:** The full quadratic form for the second-order term in the objective function is $\frac{1}{2}\bar{\boldsymbol{\alpha}}^\top H\bar{\boldsymbol{\alpha}}$, where the matrix $H$ is given by:

$$H = \frac{\mathcal{T}^2}{N}bc^\top - \frac{\mathcal{T}}{N}\sum_{l=1}^M\Psi_l(\omega)\Sigma^{(l)}$$

This matches Equation 8 in Theorem 3.3.

## C  THE PROOF OF THE CONVERGENCE

### C.1  PROOF FOR MONTE CARLO CONVERGENCE

Let the Monte Carlo estimates be defined as the average of $N$ independent and identically distributed (i.i.d.) samples, denoted by $g_h^{(i)}$ and $H_h^{(i)}$, corresponding to trajectories $\omega_i\sim p(\omega;\pi)$.

$$\nabla_h\hat{J}(h_k) = \frac{1}{N}\sum_{i=1}^N g_h^{(i)}$$

$$\nabla^2_h\hat{J}(h_k) = \frac{1}{N}\sum_{i=1}^N H_h^{(i)}$$

The true Fréchet derivatives are the expectations of these samples (we use $\mathbb{E}[\cdot]$ as shorthand for $\mathbb{E}_{\omega\sim p(\omega;\pi)}[\cdot]$):

$$\nabla_h J(h_k) = \mathbb{E}[g_h]$$
$$\nabla^2_h J(h_k) = \mathbb{E}[H_h]$$

Consider the expected squared norm for the first-order derivative estimate. Since the Monte Carlo estimator is unbiased ($\mathbb{E}[\nabla_h\hat{J}(h_k)] = \nabla_h J(h_k)$), the expected squared norm equals the variance:

$$\mathbb{E}\left[\left\|\nabla_h\hat{J}(h_k) - \nabla_h J(h_k)\right\|^2\right] = \mathrm{Var}(\nabla_h\hat{J}(h_k))$$

The variance of the mean of $N$ random variables is $1/N$ times the variance of a single variable:

$$\mathrm{Var}(\nabla_h\hat{J}(h_k)) = \mathrm{Var}\left(\frac{1}{N}\sum_{i=1}^N g_h^{(i)}\right) = \frac{1}{N^2}\sum_{i=1}^N\mathrm{Var}(g_h^{(i)}) = \frac{1}{N}\mathrm{Var}(g_h)$$

The variance of $g_h$ is defined as:

$$\mathrm{Var}(g_h) = \mathbb{E}[\|g_h - \mathbb{E}[g_h]\|^2]$$

Using the property $\mathrm{Var}(X) = \mathbb{E}[\|X\|^2] - \|\mathbb{E}[X]\|^2$ (which holds according to the definition of RKHS) and the fact that $\|\mathbb{E}[X]\|^2\geq 0$:

$$\mathrm{Var}(g_h) = \mathbb{E}[\|g_h\|^2] - \|\mathbb{E}[g_h]\|^2 \leq \mathbb{E}[\|g_h\|^2]$$

Applying the assumption $\mathbb{E}[\|g_h\|^2]\leq\sigma_0^2$:

$$\mathrm{Var}(g_h)\leq\sigma_0^2$$

Substituting this back, we find the standard convergence rate for the expected squared norm:

$$\mathbb{E}\left[\left\|\nabla_h \hat{J}(h_k) - \nabla_h J(h_k)\right\|^2\right] \le \frac{\sigma_0^2}{N}$$

Similarly, for the second-order derivative estimate:

$$\mathbb{E}\left[\left\|\nabla_h^2 \hat{J}(h_k) - \nabla_h^2 J(h_k)\right\|^2\right] = \mathrm{Var}(\nabla_h^2 \hat{J}(h_k))$$

$$\mathrm{Var}(\nabla_h^2 \hat{J}(h_k)) = \mathrm{Var}\left(\frac{1}{N}\sum_{i=1}^N H_h^{(i)}\right) = \frac{1}{N}\mathrm{Var}(H_h)$$

$$\mathrm{Var}(H_h) = \mathbb{E}[\|H_h - \mathbb{E}[H_h]\|^2] \le \mathbb{E}[\|H_h\|^2]$$

Applying the assumption $\mathbb{E}[\|H_h\|^2] \le \sigma_1^2$:

$$\mathrm{Var}(H_h) \le \sigma_1^2$$

Substituting back, the standard convergence rate for the expected squared norm is:

$$\mathbb{E}\left[\left\|\nabla_h^2 \hat{J}(h_k) - \nabla_h^2 J(h_k)\right\|^2\right] \le \frac{\sigma_1^2}{N}$$

This completes the proof.

### C.2 PROOF FOR TAYLOR UPPER BOUND

Let $\Delta h = h_2 - h_1$. Define the auxiliary function $\phi : [0, 1] \to \mathbb{R}$ by $\phi(t) = J(h_1 + t\Delta h)$. By the chain rule for Fréchet derivatives:

$$\phi'(t) = \langle \nabla_h J(h_1 + t\Delta h), \Delta h \rangle$$

$$\phi''(t) = \langle \nabla_h^2 J(h_1 + t\Delta h) \circ \Delta h, \Delta h \rangle$$

Using Taylor's theorem with integral remainder for $\phi(t)$:

$$\phi(1) = \phi(0) + \phi'(0) + \int_0^1 (1-t)\phi''(t)dt$$

Substituting the expressions for $\phi$, $\phi'$, and $\phi''$:

$$J(h_2) = J(h_1) + \langle \nabla_h J(h_1), \Delta h \rangle + \int_0^1 (1-t)\langle \nabla_h^2 J(h_1 + t\Delta h) \circ \Delta h, \Delta h \rangle dt$$

We introduce the second-order term at $h_1$. Note that $\int_0^1 (1-t)dt = 1/2$. Thus,

$$\frac{1}{2}\langle \nabla_h^2 J(h_1) \circ \Delta h, \Delta h \rangle = \int_0^1 (1-t)\langle \nabla_h^2 J(h_1) \circ \Delta h, \Delta h \rangle dt$$

Adding and subtracting this term within the integral expression for $J(h_2)$:

$$J(h_2) = J(h_1) + \langle \nabla_h J(h_1), \Delta h \rangle + \frac{1}{2}\langle \nabla_h^2 J(h_1) \circ \Delta h, \Delta h \rangle$$

$$+ \int_0^1 (1-t)\langle \nabla_h^2 J(h_1 + t\Delta h) \circ \Delta h, \Delta h \rangle dt$$

$$- \int_0^1 (1-t)\langle \nabla_h^2 J(h_1) \circ \Delta h, \Delta h \rangle dt$$

$$= J(h_1) + \langle \nabla_h J(h_1), \Delta h \rangle + \frac{1}{2}\langle \nabla_h^2 J(h_1) \circ \Delta h, \Delta h \rangle$$

$$+ \int_0^1 (1-t)\langle [\nabla_h^2 J(h_1 + t\Delta h) - \nabla_h^2 J(h_1)] \circ \Delta h, \Delta h \rangle dt$$

Let $R_2$ be the remainder term:

$$R_2 = \int_0^1 (1-t)\langle [\nabla_h^2 J(h_1 + t\Delta h) - \nabla_h^2 J(h_1)] \circ \Delta h, \Delta h \rangle dt$$

Through Cauchy-Schwarz inequality, the quadratic form $\langle Av, v \rangle \leq \|A\|\|v\|^2$:

$$\langle [\nabla_h^2 J(h_1 + t\Delta h) - \nabla_h^2 J(h_1)] \circ \Delta h, \Delta h \rangle \leq \|\nabla_h^2 J(h_1 + t\Delta h) - \nabla_h^2 J(h_1)\|\|\Delta h\|^2$$

Using the Lipschitz continuity of the Hessian in Assumption 4.1:

$$\|\nabla_h^2 J(h_1 + t\Delta h) - \nabla_h^2 J(h_1)\| \leq L\|(h_1 + t\Delta h) - h_1\| = L\|t\Delta h\| = Lt\|\Delta h\| \quad \text{(since } t \geq 0)$$

Substituting this bound into the integral for $R_2$:

$$R_2 \leq \int_0^1 (1-t)(Lt\|\Delta h\|)\|\Delta h\|^2 dt$$

$$= L\|\Delta h\|^3 \int_0^1 (1-t)t \, dt$$

$$= L\|\Delta h\|^3 \int_0^1 (t - t^2) dt$$

$$= \frac{L}{6}\|\Delta h\|^3$$

Substituting this upper bound for $R_2$ back into the expression for $J(h_2)$ and replacing $\Delta h$ with $h_2 - h_1$:

$$J(h_2) \leq J(h_1) + \langle \nabla_h J(h_1), h_2 - h_1 \rangle + \frac{1}{2}\langle \nabla_h^2 J(h_1) \circ (h_2 - h_1), h_2 - h_1 \rangle + \frac{L}{6}\|h_2 - h_1\|^3$$

This completes the proof.

## C.3 PROOF FOR THE UPPER BOUND OF THE STEP NORM

To prove this upper bound, we first need to introduce a lemma to show the optimality conditions of the iteration step:

**Lemma C.1** *(Optimality conditions) Let*

$$\Delta h = \underset{\bar{h} \in \mathcal{H}_K}{\operatorname{argmin}} \left\{ \left\langle \nabla_h \hat{J}(h_k), \bar{h} \right\rangle + \frac{1}{2}\left\langle \nabla_h^2 \hat{J}(h_k) \circ \bar{h}, \bar{h} \right\rangle + \frac{\beta}{6}\|\bar{h}\|^3 \right\}$$

*, then it satisfies that:*

$$\nabla_h \hat{J}(h_k) + \nabla_h^2 \hat{J}(h_k) \circ \Delta h + \frac{\beta}{2}\|\Delta h\|\Delta h = 0 \text{ (necessary condition)},$$

$$\langle (\nabla_h^2 \hat{J} \circ u, u \rangle + \langle \frac{\beta}{2}\|\Delta h\| I \circ u, u \rangle \geq 0 \, \forall u \in \mathcal{H}_K \text{ (sufficient condition)},$$

where $I$ is the identity operator on $\mathcal{H}_K \otimes \mathcal{H}_K \to \mathcal{H}_K$.

**Proof:** To simplify the notation in the proof, we denote $g = \nabla J(h_k) \in \mathcal{H}_K$, $H_k = \nabla^2 J(h_k)$ and the objective function $M : \mathcal{H}_K \to \mathbb{R}$:

$$M(\bar{h}) = \langle g, \bar{h} \rangle + \frac{1}{2}\langle (H_k \circ \bar{h}), \bar{h} \rangle + \frac{\beta}{6}\|\bar{h}\|^3.$$

The first Fréchet derivative of $M$ at $\bar{h}$ is given by:

$$\nabla M(\bar{h}) = g + H_k \circ \bar{h} + \frac{\beta}{2}\|\bar{h}\|\bar{h}$$

Since $\Delta h$ is a minimizer, it must satisfy the necessary condition $\nabla M(\Delta h) = 0$:

$$g + H_k \circ \Delta h + \frac{\beta}{2}\|\Delta h\|\Delta h = 0.$$

We now prove the standard second-order necessary condition for $\Delta h$.

Let $\Delta h$ be a minimizer of $M(\bar{h})$ and let $\frac{\beta}{2}\|\Delta h\| = \frac{\beta}{2}\|\Delta h\|$. Then the operator $H_k + \frac{\beta}{2}\|\Delta h\|I$ must be positive semi-definite, i.e.,

$$\langle(H_k \circ u, u\rangle + \langle\frac{\beta}{2}\|\Delta h\|I \circ u, u\rangle \geq 0 \;\rightarrow\; H_k + \frac{\beta}{2}\|\Delta h\|I \succeq 0$$

We proceed by contradiction. Assume that $H_k + \frac{\beta}{2}\|\Delta h\|I$ is not positive semi-definite ($H_k + \frac{\beta}{2}\|\Delta h\|I \not\succeq 0$). This implies that there exists a direction $u \in \mathcal{H}_K$ with $\|u\| = 1$ such that its associated quadratic form is negative:

$$\mu := \langle(H_k + \frac{\beta}{2}\|\Delta h\|I) \circ u, u\rangle < 0$$

Consider the Taylor expansion of $M$ around the minimizer $\Delta h$ along the direction $u$ for a small step $\epsilon \in \mathbb{R}$. Using Taylor's theorem in Hilbert spaces:

$$M(\Delta h + \epsilon u) = M(\Delta h) + \epsilon\langle\nabla M(\Delta h), u\rangle + \frac{\epsilon^2}{2}\langle(\nabla^2 M(\Delta h)) \circ u, u\rangle + O(\epsilon^3)$$

Since $\nabla M(\Delta h) = 0$ from the first-order condition, this simplifies to:

$$M(\Delta h + \epsilon u) = M(\Delta h) + \frac{\epsilon^2}{2}\langle(\nabla^2 M(\Delta h)) \circ u, u\rangle + O(\epsilon^3)$$

The second Fréchet derivative (Hessian operator) of $M$ at $\bar{h}$ is calculated as:

$$\nabla^2 M(\bar{h}) = H_k + \frac{\beta}{2}\frac{\bar{h} \otimes \bar{h}}{\|\bar{h}\|} + \frac{\beta}{2}\|\bar{h}\|I$$

Evaluating at $\Delta h$ (assuming $\Delta h \neq 0$, which implies $\frac{\beta}{2}\|\Delta h\| > 0$; the case $\Delta h = 0$ requires separate, simpler verification) and substituting $\frac{\beta}{2}\|\Delta h\| = \frac{\beta}{2}\|\Delta h\|$ yields:

$$\nabla^2 M(\Delta h) = H_k + \frac{\beta}{2}\frac{\Delta h \otimes \Delta h}{\|\Delta h\|} + \frac{\beta}{2}\|\Delta h\|I = (H_k + \frac{\beta}{2}\|\Delta h\|I) + \frac{\beta}{2}\frac{\Delta h \otimes \Delta h}{\|\Delta h\|}$$

Now, substitute this Hessian back into the Taylor expansion. The quadratic term is:

$$\langle(\nabla^2 M(\Delta h)) \circ u, u\rangle = \langle(H_k + \frac{\beta}{2}\|\Delta h\|I) \circ u, u\rangle + \frac{\beta}{2\|\Delta h\|}\langle\langle\Delta h, u\rangle\Delta h, u\rangle$$

Using the definition of $\mu$ and properties of the inner product, this becomes:

$$\langle(\nabla^2 M(\Delta h)) \circ u, u\rangle = \mu + \frac{\beta}{2\|\Delta h\|}\langle\Delta h, u\rangle^2$$

The Taylor expansion for the difference is thus:

$$M(\Delta h + \epsilon u) - M(\Delta h) = \frac{\epsilon^2}{2}\left(\mu + \frac{\beta}{2\|\Delta h\|}\langle\Delta h, u\rangle^2\right) + O(\epsilon^3)$$

Let $K = \mu + \frac{\beta}{2\|\Delta h\|}\langle\Delta h, u\rangle^2$. By assumption, $\mu < 0$. The second term $\frac{\beta}{2\|\Delta h\|}\langle\Delta h, u\rangle^2$ is non-negative. Since the assumption $H_k + \frac{\beta}{2}\|\Delta h\|I \not\succeq 0$ leads to a contradiction in all cases, the assumption must be false. Therefore, we must conclude that $H_k + \frac{\beta}{2}\|\Delta h\|I \succeq 0$, which complete the proof.

Now we continue the proof for the upper bound of the step norm. Through the Taylor upper bound in lemma 4.2 we know that:

$$J(h_2) \leq J(h_1) + \langle\nabla_h J(h_1), h_2 - h_1\rangle + \frac{1}{2}\langle\nabla_h^2 J(h_1) \circ (h_2 - h_1), h_2 - h_1\rangle + \frac{L}{6}\|h_2 - h_1\|^3.$$

For $\Delta h = h_2 - h_1$ that satisfies the optimality conditions, we can use the necessary condition to establish that:

$$J(h_2) \leq J(h_1) + \langle\nabla_h J(h_1) - \nabla_h \hat{J}(h_k), \Delta h\rangle + \frac{1}{2}\langle\left(\nabla_h^2 J(h_1) - \nabla_h^2 \hat{J}(h_1)\right) \circ (\Delta h), \Delta h\rangle +$$

$$\frac{L}{6}\|\Delta h\|^3 - \frac{1}{2}\langle\left(\nabla_h^2 \hat{J}(h_1)\right) \circ (\Delta h), \Delta h\rangle - \frac{\beta}{2}\|\Delta h\|^3.$$

$$(11)$$

For $L \leq \beta$ and the sufficient condition in Lemma C.1, we could find that

$$\frac{L}{6}\|\Delta h\|^3 \leq \frac{\beta}{6}\|\Delta h\|^3$$

$$-\frac{1}{2}\langle\left(\nabla_h^2 \hat{J}(h_1)\right) \circ (\Delta h), \Delta h\rangle \leq \frac{\beta}{4}\|\Delta h\|^3$$

Substituting back into the inequality 11, we can find that:

$$\frac{\beta}{12}\|\Delta h\|^3 \leq J(h_1) - J(h_2)$$

$$+ \langle\nabla_h J(h_1) - \nabla_h \hat{J}(h_k), \Delta h\rangle + \frac{1}{2}\langle\left(\nabla_h^2 J(h_1) - \nabla_h^2 \hat{J}(h_1)\right) \circ (\Delta h), \Delta h\rangle \tag{12}$$

For the gradient error term, applying Cauchy-Schwarz and Young's inequality ($ab \leq C_1 a^{3/2} + \epsilon b^3$ with $a = \|\nabla_h J(h_k) - \nabla_h \hat{J}(h_k)\|$, $b = \|\Delta h_k\|$, $p = 3/2$, $q = 3$, and $\epsilon = \beta/36$):

$$\langle\nabla_h J(h_k) - \nabla_h \hat{J}(h_k), \Delta h_k\rangle$$

$$\leq \|\nabla_h J(h_k) - \nabla_h \hat{J}(h_k)\|\|\Delta h_k\|$$

$$\leq \frac{4\sqrt{3}}{3\sqrt{\beta}}\|\nabla_h J(h_k) - \nabla_h \hat{J}(h_k)\|^{3/2} + \frac{\beta}{36}\|\Delta h_k\|^3.$$

For the Hessian error term, applying generalized Cauchy-Schwarz and Young's inequality ($cd^2 \leq C_2 c^3 + \epsilon d^3$ with $c = \frac{1}{2}\|\nabla_h^2 J(h_k) - \nabla_h^2 \hat{J}(h_k)\|$, $d = \|\Delta h_k\|$, $p = 3$, $q = 3/2$, and $\epsilon = \beta/36$):

$$\frac{1}{2}\langle(\nabla_h^2 J(h_k) - \nabla_h^2 \hat{J}(h_k)) \circ (\Delta h_k), \Delta h_k\rangle$$

$$\leq \frac{1}{2}\|\nabla_h^2 J(h_k) - \nabla_h^2 \hat{J}(h_k)\|\|\Delta h_k\|^2$$

$$\leq \frac{24}{\beta^2}\|\nabla_h^2 J(h_k) - \nabla_h^2 \hat{J}(h_k)\|^3 + \frac{\beta}{36}\|\Delta h_k\|^3.$$

Substituting these bounds back into 12:

$$\frac{\beta}{12}\|\Delta h_k\|^3 \leq J(h_k) - J(h_{k+1})$$

$$+ \frac{4\sqrt{3}}{3\sqrt{\beta}}\|\nabla_h J(h_k) - \nabla_h \hat{J}(h_k)\|^{3/2} + \frac{\beta}{36}\|\Delta h_k\|^3$$

$$+ \frac{24}{\beta^2}\|\nabla_h^2 J(h_k) - \nabla_h^2 \hat{J}(h_k)\|^3 + \frac{\beta}{36}\|\Delta h_k\|^3.$$

Rearranging terms yields:

$$\left(\frac{\beta}{12} - \frac{\beta}{36} - \frac{\beta}{36}\right)\|\Delta h_k\|^3 \leq J(h_k) - J(h_{k+1}) +$$

$$\frac{4\sqrt{3}}{3\sqrt{\beta}}\|\nabla_h J(h_k) - \nabla_h \hat{J}(h_k)\|^{3/2} + \frac{24}{\beta^2}\|\nabla_h^2 J(h_k) - \nabla_h^2 \hat{J}(h_k)\|^3,$$

which simplifies to:

$$\frac{\beta}{36}\|\Delta h_k\|^3 \leq J(h_k) - J(h_{k+1}) + \frac{4\sqrt{3}}{3\sqrt{\beta}}\|\nabla_h J(h_k) - \nabla_h \hat{J}(h_k)\|^{3/2} + \frac{24}{\beta^2}\|\nabla_h^2 J(h_k) - \nabla_h^2 \hat{J}(h_k)\|^3. \tag{13}$$

Now, we take the total expectation $\mathbb{E}[\cdot]$ over all randomness. Using Lemma C.1 and properties of expectation (Jensen's inequality), we bound the expected error terms:

$$\mathbb{E}\left[\|\nabla_h J(h_k) - \nabla_h \hat{J}(h_k)\|^{3/2}\right] \leq \left(\mathbb{E}\left[\|\nabla_h J(h_k) - \nabla_h \hat{J}(h_k)\|^2\right]\right)^{3/4} \leq \left(\frac{\sigma_0^2}{N}\right)^{3/4} = \frac{\sigma_0^{3/2}}{N^{3/4}}.$$

$$\mathbb{E}\left[\|\nabla_h^2 J(h_k) - \nabla_h^2 \hat{J}(h_k)\|^3\right] \leq \left(\frac{\sigma_1}{\sqrt{N}}\right)^3 = \frac{\sigma_1^3}{N^{3/2}}. \quad \text{(See note below)}$$

Substituting these bounds into the expectation of 13:

$$\frac{\beta}{36}\mathbb{E}\left[\|\Delta h_k\|^3\right] \leq \mathbb{E}[J(h_k)] - \mathbb{E}[J(h_{k+1})]$$
$$+ \frac{4\sqrt{3}}{3\sqrt{\beta}}\left(\frac{\sigma_0^{3/2}}{N^{3/4}}\right) + \frac{24}{\beta^2}\left(\frac{\sigma_1^3}{N^{3/2}}\right).$$

Summing this inequality over the total iterations $k = 1, \ldots, M$:

$$\frac{\beta}{36}\sum_{k=1}^{M}\mathbb{E}\left[\|h_{k+1} - h_k\|^3\right] \leq \sum_{k=1}^{M}\left(\mathbb{E}[J(h_k)] - \mathbb{E}[J(h_{k+1})]\right)$$
$$+ \sum_{k=1}^{M}\left(\frac{4\sqrt{3}}{3\sqrt{\beta}}\frac{\sigma_0^{3/2}}{N^{3/4}} + \frac{24}{\beta^2}\frac{\sigma_1^3}{N^{3/2}}\right).$$

The first sum on the right-hand side telescopes to $\mathbb{E}[J(h_1)] - \mathbb{E}[J(h_{M+1})]$. Assuming $h_1$ is deterministic and $J(h) \geq J^*$ for some minimum value $J^*$, this sum is bounded by $J(h_1) - J^*$. The second sum consists of terms independent of the summation index $k$:

$$\sum_{k=1}^{M}(\ldots) = M\left(\frac{4\sqrt{3}}{3\sqrt{\beta}}\frac{\sigma_0^{3/2}}{N^{3/4}} + \frac{24}{\beta^2}\frac{\sigma_1^3}{N^{3/2}}\right).$$

Combining these results:

$$\frac{\beta}{36}\sum_{k=1}^{M}\mathbb{E}\left[\|h_{k+1} - h_k\|^3\right] \leq J(h_1) - J^* + M\left(\frac{4\sqrt{3}}{3\sqrt{\beta}}\frac{\sigma_0^{3/2}}{N^{3/4}} + \frac{24}{\beta^2}\frac{\sigma_1^3}{N^{3/2}}\right).$$

Let $R$ be a random variable uniformly distributed on $\{1, \ldots, M\}$, such that $P(R = k) = 1/M$. Then $\mathbb{E}[\|h_{R+1} - h_R\|^3] = \frac{1}{M}\sum_{k=1}^{M}\mathbb{E}\left[\|h_{k+1} - h_k\|^3\right]$. Dividing the inequality by $M$:

$$\frac{\beta}{36}\mathbb{E}\left[\|h_{R+1} - h_R\|^3\right] \leq \frac{J(h_1) - J^*}{M} + \left(\frac{4\sqrt{3}}{3\sqrt{\beta}}\frac{\sigma_0^{3/2}}{N^{3/4}} + \frac{24}{\beta^2}\frac{\sigma_1^3}{N^{3/2}}\right).$$

Finally, multiplying by $36/\beta$ isolates the expected cubic step norm for a randomly chosen iteration $R$:

$$\mathbb{E}\left[\|h_{R+1} - h_R\|^3\right] \leq \frac{36(J(h_1) - J^*)}{\beta M} + \frac{36}{\beta}\left(\frac{4\sqrt{3}}{3\sqrt{\beta}}\frac{\sigma_0^{3/2}}{N^{3/4}} + \frac{24}{\beta^2}\frac{\sigma_1^3}{N^{3/2}}\right)$$
$$\leq \frac{36(J(h_1) - J^*)}{\beta M} + \frac{48\sqrt{3}}{\beta^{3/2}}\frac{\sigma_0^{3/2}}{N^{3/4}} + \frac{864}{\beta^3}\frac{\sigma_1^3}{N^{3/2}}. \tag{14}$$

We can box the final result for emphasis:

$$\mathbb{E}\left[\|h_{R+1} - h_R\|^3\right] \leq \frac{36(J(h_1) - J^*)}{\beta M} + \frac{48\sqrt{3}}{\beta^{3/2}}\frac{\sigma_0^{3/2}}{N^{3/4}} + \frac{864}{\beta^3}\frac{\sigma_1^3}{N^{3/2}}.$$

This completes the derivation of the upper bound on the expected cubic step norm.

## C.4 PROOF FOR THE LOWER BOUND OF THE STEP NORM

From the proof of the Taylor upper bound in Appendix C.2, we could similarly derive the first-order Taylor upper bound as

$$\nabla J(h_2) \leq \nabla J(h_1) + \langle \nabla_h^2 J(h_1) \circ (h_2 - h_1), h_2 - h_1 \rangle + \frac{L}{2}\|h_2 - h_1\|^3. \tag{15}$$

Through this, we could prove the lower bound by first constructing this auxiliary equation through the optimality conditions in Lemma C.1:

$$\nabla J(h_{k+1}) = \nabla J(h_{k+1}) - (\nabla_h \hat{J}(h_k) + \nabla_h^2 \hat{J}(h_k) \circ \Delta h_k + \frac{\beta}{2}\|\Delta h_k\|\Delta h_k)$$

$$= \left[\nabla J(h_{k+1}) - \nabla J(h_k) - \nabla^2 J(h_k) \circ \Delta h_k\right] \quad \text{(Term 1)}$$

$$+ \left[\nabla J(h_k) - \nabla_h \hat{J}(h_k)\right] \quad \text{(Term 2)}$$

$$+ \left[(\nabla^2 J(h_k) - \nabla_h^2 \hat{J}(h_k)) \circ \Delta h_k\right] \quad \text{(Term 3)}$$

$$- \frac{\beta}{2}\|\Delta h_k\|\Delta h_k \quad \text{(Term 4)}$$

Taking norms and applying the triangle inequality:

$$\|\nabla J(h_{k+1})\| \leq \|\nabla J(h_{k+1}) - \nabla J(h_k) - \nabla^2 J(h_k) \circ \Delta h_k\|$$

$$+ \|\nabla J(h_k) - \nabla_h \hat{J}(h_k)\|$$

$$+ \|(\nabla^2 J(h_k) - \nabla_h^2 \hat{J}(h_k)) \circ \Delta h_k\|$$

$$+ \|-\frac{\beta}{2}\|\|\Delta h_k\|\Delta h_k$$

We bound the terms using Assumption 4.1 for Term 1, norm properties for Term 3, and direct calculation for Term 4:

$$\|\nabla J(h_{k+1})\| \leq \frac{L}{2}\|\Delta h_k\|^2 + \|\nabla J(h_k) - \nabla_h \hat{J}(h_k)\|$$

$$+ \|\nabla^2 J(h_k) - \nabla_h^2 \hat{J}(h_k)\|\|\Delta h_k\| + \frac{\beta}{2}\|\Delta h_k\|^2$$

Applying Young's inequality $ab \leq \frac{a^2}{2C} + \frac{Cb^2}{2}$ with $C = L + \beta$ to the term involving the Hessian error:

$$\|\nabla^2 J(h_k) - \nabla_h^2 \hat{J}(h_k)\|\|\Delta h_k\| \leq \frac{\|\nabla^2 J(h_k) - \nabla_h^2 \hat{J}(h_k)\|^2}{2(L+\beta)} + \frac{(L+\beta)\|\Delta h_k\|^2}{2}$$

Substituting this back and collecting terms with $\|\Delta h_k\|^2$:

$$\|\nabla J(h_{k+1})\| \leq \left(\frac{L}{2} + \frac{L+\beta}{2} + \frac{\beta}{2}\right)\|\Delta h_k\|^2$$

$$+ \|\nabla J(h_k) - \nabla_h \hat{J}(h_k)\| + \frac{\|\nabla^2 J(h_k) - \nabla_h^2 \hat{J}(h_k)\|^2}{2(L+\beta)}$$

$$= (L+\beta)\|\Delta h_k\|^2$$

$$+ \|\nabla J(h_k) - \nabla_h \hat{J}(h_k)\| + \frac{\|\nabla^2 J(h_k) - \nabla_h^2 \hat{J}(h_k)\|^2}{2(L+\beta)}$$

Now, take the total expectation $\mathbb{E}[\cdot]$. Using Lemma C.1 and Jensen's inequality:

$$\mathbb{E}\left[\|\nabla J(h_k) - \nabla_h \hat{J}(h_k)\|\right] \leq \sqrt{\mathbb{E}\left[\|\nabla J(h_k) - \nabla_h \hat{J}(h_k)\|^2\right]} \leq \frac{\sigma_0}{\sqrt{N}}$$

$$\mathbb{E}\left[\|\nabla^2 J(h_k) - \nabla_h^2 \hat{J}(h_k)\|^2\right] \leq \frac{\sigma_1^2}{N}$$

Applying expectation to the inequality for $\|\nabla J(h_{k+1})\|$:

$$\mathbb{E}[\|\nabla J(h_{k+1})\|] \leq (L+\beta)\mathbb{E}[\|\Delta h_k\|^2]$$

$$+ \mathbb{E}\left[\|\nabla J(h_k) - \nabla_h \hat{J}(h_k)\|\right] + \frac{\mathbb{E}\left[\|\nabla^2 J(h_k) - \nabla_h^2 \hat{J}(h_k)\|^2\right]}{2(L+\beta)}$$

$$\leq (L+\beta)\mathbb{E}[\|\Delta h_k\|^2] + \frac{\sigma_0}{\sqrt{N}} + \frac{\sigma_1^2}{2N(L+\beta)}$$

Rearranging to isolate the expected squared step norm:

$$(L + \beta)\, \mathbb{E}[\|\Delta h_k\|^2] \geq \mathbb{E}[\|\nabla J(h_{k+1})\|] - \frac{\sigma_0}{\sqrt{N}} - \frac{\sigma_1^2}{2N(L+\beta)}$$

$$\mathbb{E}[\|h_{k+1} - h_k\|^2] \geq \frac{1}{L+\beta}\left(\mathbb{E}[\|\nabla J(h_{k+1})\|] - \frac{\sigma_0}{\sqrt{N}} - \frac{\sigma_1^2}{2N(L+\beta)}\right)$$

This completes the proof for the lower bound based on the gradient norm, using the corrected regularization term.

### C.5 PROOF FOR CONVERGENCE THEOREM

From Lemma 4.3, the upper bound on the expected cubic step norm for $R \sim \text{Uniform}\{1, \ldots, M\}$ is:

$$\mathbb{E}\left[\|h_{R+1} - h_R\|^3\right] \leq \frac{36(J(h_1) - J^*)}{\beta M} + \frac{48\sqrt{3}}{\beta^{3/2}}\frac{\sigma_0^{3/2}}{N^{3/4}} + \frac{864}{\beta^3}\frac{\sigma_1^3}{N^{3/2}}.$$

As the number of iterations $M \to \infty$ and the batch size $N \to \infty$, the right-hand side approaches zero. Thus,

$$\lim_{M,N\to\infty} \mathbb{E}\left[\|h_{R+1} - h_R\|^3\right] = 0. \tag{16}$$

Using Lyapunov's inequality, $\mathbb{E}[\|h_{R+1} - h_R\|^2] \leq (\mathbb{E}[\|h_{R+1} - h_R\|^3])^{2/3}$. Taking the limit as $M, N \to \infty$ and using 16:

$$\lim_{M,N\to\infty} \mathbb{E}\left[\|h_{R+1} - h_R\|^2\right] = 0. \tag{17}$$

From Lemma 4.4, we rearrange the inequality which holds for any iteration $k$:

$$\mathbb{E}[\|\nabla J(h_{k+1})\|] \leq (L + \beta)\mathbb{E}[\|h_{k+1} - h_k\|^2] + \frac{\sigma_0}{\sqrt{N}} + \frac{\sigma_1^2}{2N(L+\beta)}.$$

Now, we take the expectation over the random index $R \sim \text{Uniform}\{1, \ldots, M\}$. Since $R$ selects one of the iterations $k \in \{1, \ldots, M\}$ uniformly, taking the expectation of the inequality with respect to $R$ effectively averages it:

$$\mathbb{E}_R\left[\mathbb{E}[\|\nabla J(h_{R+1})\|]\right] \leq \mathbb{E}_R\left[(L + \beta)\mathbb{E}[\|h_{R+1} - h_R\|^2] + \frac{\sigma_0}{\sqrt{N}} + \frac{\sigma_1^2}{2N(L+\beta)}\right].$$

Here, $\mathbb{E}_R$ denotes the expectation over the choice of $R$. Let $\mathbb{E}[\cdot]$ denote the total expectation (over the process history and $R$). The inequality becomes:

$$\mathbb{E}[\|\nabla J(h_{R+1})\|] \leq (L + \beta)\mathbb{E}[\|h_{R+1} - h_R\|^2] + \frac{\sigma_0}{\sqrt{N}} + \frac{\sigma_1^2}{2N(L+\beta)}.$$

Taking the limit as $M \to \infty$ and $N \to \infty$:

$$\lim_{M,N\to\infty} \mathbb{E}[\|\nabla J(h_{R+1})\|] \leq \lim_{M,N\to\infty}\left((L + \beta)\mathbb{E}[\|h_{R+1} - h_R\|^2] + \frac{\sigma_0}{\sqrt{N}} + \frac{\sigma_1^2}{2N(L+\beta)}\right)$$

$$= (L + \beta)\lim_{M,N\to\infty} \mathbb{E}[\|h_{R+1} - h_R\|^2] + \lim_{N\to\infty}\frac{\sigma_0}{\sqrt{N}} + \lim_{N\to\infty}\frac{\sigma_1^2}{2N(L+\beta)}$$

$$= (L + \beta) \times 0 + 0 + 0 \quad \text{(Using 17)}$$

$$= 0.$$

Since $\mathbb{E}[\|\nabla J(h_{R+1})\|] \geq 0$, we conclude that:

$$\lim_{M,N\to\infty} \mathbb{E}[\|\nabla J(h_{R+1})\|] = 0.$$

As $\mathbb{E}[\|\nabla J(h_R)\|]$ differs from $\mathbb{E}[\|\nabla J(h_{R+1})\|]$ by terms that vanish as $M \to \infty$ (typically $\frac{1}{M}(\mathbb{E}[\|\nabla J(h_{M+1})\|] - \mathbb{E}[\|\nabla J(h_1)\|])$), we can equivalently state:

$$\lim_{M,N\to\infty} \mathbb{E}[\|\nabla J(h_R)\|] = 0.$$

This proves that the expected gradient norm at a randomly chosen iteration converges to zero.

## D    PROOF FOR QUADRATIC CONVERGENCE

Let the error at iteration $k$ be $e_k = h_k - h^*$. The update gives $h_{k+1} - h^* = h_k - h^* + \Delta h_k$, so $e_{k+1} = e_k + \Delta h_k$. The update step satisfies the optimal condition C.1, which is given by:

$$\nabla J(h_k) + \nabla^2 J(h_k) \circ \Delta h_k + \frac{\beta}{2}\|\Delta h_k\|\Delta h_k = 0.$$

Rearranging yields:

$$(\nabla^2 J(h_k) + \frac{\beta}{2}\|\Delta h_k\|\mathcal{I}) \circ \Delta h_k = -\nabla J(h_k)$$

where $\mathcal{I}$ is the identity operator.

We expand $\nabla J(h_k)$ around $h^*$ using Taylor's theorem (similar to the derivation in Appendix C.2):

$$\nabla J(h_k) = \nabla J(h^*) + \nabla^2 J(h^*) \circ (h_k - h^*) + \mathcal{R}_1(h_k, h^*)$$

where $\nabla J(h^*) = 0$ and the remainder term satisfies $\|\mathcal{R}_1(h_k, h^*)\| \leq \frac{L}{6}\|h_k - h^*\|^2 = \frac{L}{6}\|e_k\|^2$ for some constant $\frac{L}{6}$ when $h_k$ is near $h^*$. Thus,

$$\nabla J(h_k) = \nabla^2 J(h^*) \circ e_k + \mathcal{R}_1(h_k, h^*)$$

Substitute this into the rearranged update equation:

$$(\nabla^2 J(h_k) + \frac{\beta}{2}\|\Delta h_k\|\mathcal{I}) \circ \Delta h_k = -\nabla^2 J(h^*) \circ e_k - \mathcal{R}_1(h_k, h^*)$$

Let $H_k = \nabla^2 J(h_k)$ and $H^* = \nabla^2 J(h^*)$. Let $A_k = H_k + \frac{\beta}{2}\|\Delta h_k\|\mathcal{I}$. The equation is $A_k \circ \Delta h_k = -H^* \circ e_k - \mathcal{R}_1(h_k, h^*)$. Assuming the norm of the inverse operator $\|A_k^{-1}\|_{op}$ will be bounded by some constant $B$, and we assume that the update step is sufficiently small that $\|\Delta h_k\| \leq L\|e_k\|$.

Now, substitute $\Delta h_k = e_{k+1} - e_k$ into the equation $A_k \circ \Delta h_k = -H^* \circ e_k - \mathcal{R}_1(h_k, h^*)$:

$$A_k \circ (e_{k+1} - e_k) = -H^* \circ e_k - \mathcal{R}_1(h_k, h^*)$$

$$A_k \circ e_{k+1} = A_k \circ e_k - H^* \circ e_k - \mathcal{R}_1(h_k, h^*)$$

$$A_k \circ e_{k+1} = (H_k + \frac{\beta}{2}\|\Delta h_k\|\mathcal{I} - H^*) \circ e_k - \mathcal{R}_1(h_k, h^*)$$

$$A_k \circ e_{k+1} = (H_k - H^*) \circ e_k + \frac{\beta}{2}\|\Delta h_k\|e_k - \mathcal{R}_1(h_k, h^*)$$

Applying the inverse $A_k^{-1}$:

$$e_{k+1} = A_k^{-1} \circ \left[ (H_k - H^*) \circ e_k + \frac{\beta}{2}\|\Delta h_k\|e_k - \mathcal{R}_1(h_k, h^*) \right]$$

Taking norms and using the triangle inequality:

$$\|e_{k+1}\| \leq \|A_k^{-1}\|_{op} \left\| (H_k - H^*) \circ e_k + \frac{\beta}{2}\|\Delta h_k\|e_k - \mathcal{R}_1(h_k, h^*) \right\|$$

$$\leq \|A_k^{-1}\|_{op} \left( \|H_k - H^*\|_{op}\|e_k\| + \frac{\beta}{2}\|\Delta h_k\|\|e_k\| + \|\mathcal{R}_1(h_k, h^*)\| \right)$$

Substitute the bounds: $\|A_k^{-1}\|_{op} \leq B$, $\|H_k - H^*\|_{op} \leq L\|e_k\|$, $\|\Delta h_k\| \leq L\|e_k\|$, and $\|\mathcal{R}_1(h_k, h^*)\| \leq \frac{L}{6}\|e_k\|^2$.

$$\|e_{k+1}\| \leq B \left( (L\|e_k\|)\|e_k\| + \frac{\beta}{2}(L\|e_k\|)\|e_k\| + \frac{L}{6}\|e_k\|^2 \right)$$

$$\leq B \left( L\|e_k\|^2 + \frac{\beta L}{2}\|e_k\|^2 + \frac{L}{6}\|e_k\|^2 \right)$$

$$\leq B \left( L + \frac{\beta L}{2} + \frac{L}{6} \right) \|e_k\|^2$$

Setting $C_q = B(L + \frac{\beta L}{2} + \frac{L}{6})$, which is a positive constant independent of $k$, we have shown that

$$\|h_{k+1} - h^*\| \leq C_q\|h_k - h^*\|^2$$

This demonstrates local quadratic convergence for the deterministic version of the algorithm, provided $h_k$ is sufficiently close to $h^*$.

# E  VALIDITY OF THE ASSUMPTIONS

## E.1  VALIDITY OF ASSUMPTION 4.1 (LIPSCHITZ CONTINUITY OF THE HESSIAN)

**Claim.** Under mild regularity conditions standard in second–order analysis (Nesterov & Polyak, 2006b; Nocedal & Wright, 2006b) and in Lipschitz MDPs (Pirotta et al., 2015), the RKHS Hessian operator $\nabla_h^2 J(h)$ is Lipschitz on a neighborhood $\mathcal{N} \subset \mathcal{H}_K$; i.e., there exists $L > 0$ such that

$$\left\| \nabla_h^2 J(h_1) - \nabla_h^2 J(h_2) \right\|_{\mathrm{op}} \le L \left\| h_1 - h_2 \right\| \qquad \forall h_1, h_2 \in \mathcal{N}.$$

**Sufficient conditions.** It suffices that the following hold:

1. *Bounded horizon and rewards.* Finite horizon $T < \infty$ (as in our experiments) or a discounted infinite horizon with $\gamma < 1$, and $|r(s,a)| \le R_{\max}$.

2. *Lipschitz MDP.* The transition and reward are Lipschitz in a metric $d$ (cf. (Pirotta et al., 2015)):
   $W_1(P(\cdot|s,a), P(\cdot|s',a')) \le L_P\, d((s,a),(s',a'))$, and $|r(s,a) - r(s',a')| \le L_r\, d((s,a),(s',a'))$.

3. *Kernel boundedness and smoothness.* The kernel sections satisfy $\|K((s,a),\cdot)\| \le \kappa$ and the map $(s,a) \mapsto K((s,a),\cdot)$ is Lipschitz in RKHS norm with constant $L_K$.

4. *Softmax RKHS policy with Lipschitz log-policy.* For $\pi_h(a|s) \propto \exp(\mathcal{T}h(s,a))$ with finite $\mathcal{T}$ and $h$ restricted to $\|h\| \le B$, the log-policy and its first two Fréchet derivatives are Lipschitz in $h$.

5. *Uniform integrability.* The trajectory weights $\Psi_t(\omega)$ admit finite moments ensuring interchange of expectation and differentiation (dominated convergence).

**Sketch.** By Lemma 3.2,

$$\nabla_h^2 J(h) = \mathbb{E}\left[ \left( \sum_t \Psi_t\, \nabla_h \log \pi_h^t \right) \otimes \left( \sum_{t'} \nabla_h^\top \log \pi_h^{t'} \right) \right] - \mathbb{E}\left[ \sum_t \Psi_t\, \mathcal{T}\, \mathrm{Cov}_{a' \sim \pi_h(\cdot|s_t)}\big[K((s_t,a'),\cdot)\big] \right].$$

For the outer-product term, $\nabla_h \log \pi_h(a|s) = \mathcal{T}(K((s,a),\cdot) - \mathbb{E}_{a'\sim\pi_h}[K((s,a'),\cdot)])$. Because $(s,a) \mapsto K((s,a),\cdot)$ is Lipschitz (Assumption 3) and $h \mapsto \pi_h$ is smooth and Lipschitz on $\|h\| \le B$, the map $h \mapsto \nabla_h \log \pi_h$ is Lipschitz; bilinearity of the outer product then yields a Lipschitz bound on the first expectation. For the covariance term, the maps $h \mapsto \mathbb{E}_{a'\sim\pi_h}[K((s,a'),\cdot)]$ and $h \mapsto \mathbb{E}_{a'\sim\pi_h}[K((s,a'),\cdot) \otimes K((s,a'),\cdot)]$ are Lipschitz by the same reasoning, hence $h \mapsto \Sigma_h(s) = \mathrm{Cov}_{a'\sim\pi_h(\cdot|s)}[K((s,a'),\cdot)]$ is Lipschitz. Multiplying by bounded $|\Psi_t|$ and taking expectations over bounded-horizon (or discounted) trajectories preserves Lipschitzness. Therefore there exists $L > 0$ such that $\|\nabla_h^2 J(h_1) - \nabla_h^2 J(h_2)\|_{\mathrm{op}} \le L \|h_1 - h_2\|$. This mirrors the Lipschitz-gradient results for value functions in Lipschitz MDPs (Pirotta et al., 2015) and the smoothness assumptions used in cubic-regularized/Newton methods (Nesterov & Polyak, 2006b; Nocedal & Wright, 2006b).

## E.2  VALIDITY OF THE ASSUMPTIONS IN THEOREM 4.6 (LOCAL QUADRATIC CONVERGENCE)

The theorem rests on three local assumptions. Below we give standard sufficient conditions and why they hold in our setting.

**B.1 Bounded inverse of the regularized Hessian.** We require

$$\left\| (\nabla_h^2 J(h_k) + \tfrac{\beta}{2}\|\Delta h_k\|\mathcal{I})^{-1} \right\| \le B.$$

If $h^*$ is a strict local minimizer, then $H^* = \nabla_h^2 J(h^*)$ is positive definite. By continuity (Assumption 4.1), there exists a neighborhood $\mathcal{N}$ that $\lambda_{\min}(\nabla_h^2 J(h) + \tfrac{\beta}{2}\|\Delta h\|\mathcal{I}) \ge \mu$, so the inverse is uniformly bounded by $B \le 1/\mu$. The cubic regularizer only enlarges the spectrum, a standard safeguard in cubic-regularized Newton analyses (Nesterov & Polyak, 2006b; Nocedal & Wright, 2006b).

**B.2 Step–error proportionality** $\|\Delta h_k\| \leq K\|e_k\|$. Let $e_k = h_k - h^*$. The step solves $(\nabla_h^2 J(h_k) + \frac{\beta}{2}\|\Delta h_k\|\mathcal{I})\Delta h_k = -\nabla_h J(h_k)$. By Taylor's theorem with Lipschitz Hessian (Assumption 4.1), $\nabla_h J(h_k) = \nabla_h^2 J(h^*) e_k + r_k$ with $\|r_k\| \leq c\|e_k\|^2$. Multiplying by the bounded inverse from B.1 gives

$$\|\Delta h_k\| \leq \|A_k^{-1}\| (\|H^*\| \|e_k\| + \|r_k\|) \leq B(\|H^*\| + c\|e_k\|)\|e_k\| \leq K\|e_k\|$$

for $\|e_k\|$ small enough, establishing the proportionality in the local region.

**B.3 Initialization inside the local basin.** Quadratic convergence for Newton-type methods is inherently *local*: starting sufficiently close to $h^*$ ensures the Newton map is a contraction and that iterates remain in $\mathcal{N}$. This basin-of-attraction requirement is standard (Nocedal & Wright, 2006b) and is precisely the regime where cubic regularization attains its classical rates (Nesterov & Polyak, 2006b).

**B.4 Deterministic scope of Section 4.3.** Rates for *stochastic* Newton-type methods depend on curvature, noise, and step-size policies and typically require separate concentration and bias–variance controls; see, e.g., (Boyer & Godichon-Baggioni, 2023; Bottou et al., 2018). Our Section 4.3 therefore establishes the baseline local quadratic rate in the deterministic setting—the hallmark behavior of Newton's method—providing a principled rationale for a second-order approach to RKHS policies and a foundation for future stochastic analysis.

# F ASSET ALLOCATION EXPERIMENT

In our investment planning MDP, we formulate a state-action framework that models investment decisions under varying market conditions and resource constraints. This model captures the fundamental trade-offs between risk and return across different market states while accounting for resource dynamics.

**The state** $s \in \mathcal{S}$ is characterized by a tuple $(r, m)$ where:

- $r \in \{0, 1, \ldots, R_{max} - 1\}$ represents the discrete resource level, with $R_{max}$ being the maximum possible resource level
- $m \in \{0, 1, 2\}$ corresponds to market conditions (recession, stability, and prosperity, respectively)

The cardinality of the state space is $|\mathcal{S}| = R_{max} \times 3$.

**The action space** $\mathcal{A}$ comprises three distinct investment strategies:

- $a = 0$: Conservative investment (low risk/low return)
- $a = 1$: Balanced investment (moderate risk/moderate return)
- $a = 2$: Aggressive investment (high risk/high return)

**The state transition function** $P(s_{t+1} \mid s_t, a_t)$ models the stochastic evolution of both resource levels and market conditions:

1. Resource Dynamics: The probability of resource level transitions depends on the chosen action:
    - Conservative strategy: $P(r_{t+1}|r_t, a_t = 0) = [0.1, 0.8, 0.1, 0.0, 0.0]$ for $\Delta r \in \{-1, 0, +1, +2, +3\}$
    - Balanced strategy: $P(r_{t+1}|r_t, a_t = 1) = [0.2, 0.2, 0.4, 0.2, 0.0]$ for $\Delta r \in \{-1, 0, +1, +2, +3\}$
    - Aggressive strategy: $P(r_{t+1}|r_t, a_t = 2) = [0.4, 0.1, 0.1, 0.2, 0.2]$ for $\Delta r \in \{-1, 0, +1, +2, +3\}$
2. Market Dynamics: Market state transitions follow a Markov chain with the following probabilities:
    - Recession: $P(m_{t+1}|m_t = 0) = [0.6, 0.3, 0.1]$ for $m_{t+1} \in \{0, 1, 2\}$

- Stability: $P(m_{t+1}|m_t = 1) = [0.3, 0.4, 0.3]$ for $m_{t+1} \in \{0, 1, 2\}$
- Prosperity: $P(m_{t+1}|m_t = 2) = [0.1, 0.3, 0.6]$ for $m_{t+1} \in \{0, 1, 2\}$

The joint transition probability is computed as:

$$P((r_{t+1}, m_{t+1})|(r_t, m_t), a_t) = P(r_{t+1}|r_t, a_t) \cdot P(m_{t+1}|m_t)$$

**The reward function** $r(s_t, a_t)$ captures the expected immediate return for taking action $a_t$ in state $s_t = (r_t, m_t)$:

$$r((r_t, m_t), a_t) = B(m_t, a_t) \cdot \frac{r_t + 1}{R_{max}}$$

where $B(m_t, a_t)$ is the base reward that depends on the market state and chosen action:

- Conservative strategy ($a_t = 0$):
    - Base reward $= 1.0$ for all market states, except
    - Base reward $= 0.5$ in prosperity ($m_t = 2$) to represent opportunity cost
- Balanced strategy ($a_t = 1$):
    - Base reward $= 0.5$ in recession ($m_t = 0$)
    - Base reward $= 2.0$ in stability ($m_t = 1$)
    - Base reward $= 1.5$ in prosperity ($m_t = 2$)
- Aggressive strategy ($a_t = 2$):
    - Base reward $= -1.0$ in recession ($m_t = 0$)
    - Base reward $= 1.0$ in stability ($m_t = 1$)
    - Base reward $= 3.0$ in prosperity ($m_t = 2$)

The resource scaling factor $\frac{r_t+1}{R_{max}}$ ensures that higher resource levels amplify rewards.

**The initial state distribution** $\rho(s_0)$ is typically set to start with a medium resource level and a randomly selected market state:

$$\rho((r_0 = \lfloor R_{max}/2 \rfloor, m_0 = m)) = \frac{1}{3} \text{ for } m \in \{0, 1, 2\}$$

In our experiment, $R_{max}$ is set as 5, balancing sufficient environmental complexity with simplicity for visualization and optimal policy calculation.

## G  VISUALIZATION OF POLICY LANDSCAPES

To visualize the optimization behavior of different algorithms in Figure 1(b), we project the high-dimensional policy space onto a two-dimensional plane and overlay an approximate value landscape and the optimization trajectories.

**PCA projection of policy space.**  Let $\boldsymbol{\theta}_t^{(m)} \in \mathbb{R}^D$ denote the flattened policy parameter vector of method $m$ at iteration $t$, and let $\boldsymbol{\theta}^*$ be the analytically known optimal policy in the toy asset allocation setting. We collect all visited policies across all methods,

$$\mathcal{D} = \left\{ \boldsymbol{\theta}_t^{(m)} \,\middle|\, \forall m, \forall t \right\} \cup \left\{ \boldsymbol{\theta}^* \right\},$$

and form the data matrix $\mathbf{X} \in \mathbb{R}^{N \times D}$ with $N = |\mathcal{D}|$. We apply PCA to $\mathbf{X}$, obtain the first two principal components $\mathbf{w}_1, \mathbf{w}_2 \in \mathbb{R}^D$, and project any policy $\boldsymbol{\theta}$ to

$$\mathbf{z}(\boldsymbol{\theta}) = \left[ \mathbf{w}_1^\top (\boldsymbol{\theta} - \bar{\boldsymbol{\theta}}), \, \mathbf{w}_2^\top (\boldsymbol{\theta} - \bar{\boldsymbol{\theta}}) \right] \in \mathbb{R}^2,$$

where $\bar{\boldsymbol{\theta}}$ is the empirical mean of $\mathcal{D}$. This 2D plane captures the dominant variation of all policies and makes their relative positions (initial policies, intermediate iterates, and the optimum) directly comparable.

**Value surface reconstruction.** Let $\mathbf{z}_i = \mathbf{z}(\boldsymbol{\theta}_i)$ be the 2D projection of a policy $\boldsymbol{\theta}_i \in \mathcal{D}$ and $v_i = J(\boldsymbol{\theta}_i)$ its corresponding expected return. We construct a dense grid over the convex hull of $\{\mathbf{z}_i\}_{i=1}^N$ and approximate the value landscape by interpolating the pairs $\{(\mathbf{z}_i, v_i)\}_{i=1}^N$ using standard scattered-data interpolation (cubic interpolation when possible, and linear interpolation in sparse regions). This yields a smooth surrogate $\hat{J}(\mathbf{z})$ whose level sets define the background contour in Figure 1(b).

**Trajectory smoothing.** For each method $m$, the projected iterates $\{\mathbf{z}(\boldsymbol{\theta}_t^{(m)})\}_{t=0}^T$ are connected into a smooth curve using B-spline interpolation with a small smoothness penalty. This reduces visual clutter due to small step-to-step fluctuations while preserving the global optimization trend towards $\mathbf{z}(\boldsymbol{\theta}^*)$, which is marked as the optimal policy in Figure 1(b).

**Additional high-contrast visualization.** The main-panel Figure 1(b) was primarily designed to highlight the geometric differences between optimization trajectories and their distances to the optimal policy in the PCA-projected space. As noted by the reviewer, reward differences in the high-return region can appear visually subtle due to the relatively flat landscape around the optimum. To make these differences more apparent, we provide in Figure 3 an additional visualization with an adjusted color scale (and identical trajectories), which enhances the contrast of the reward values while keeping the underlying optimization paths unchanged.

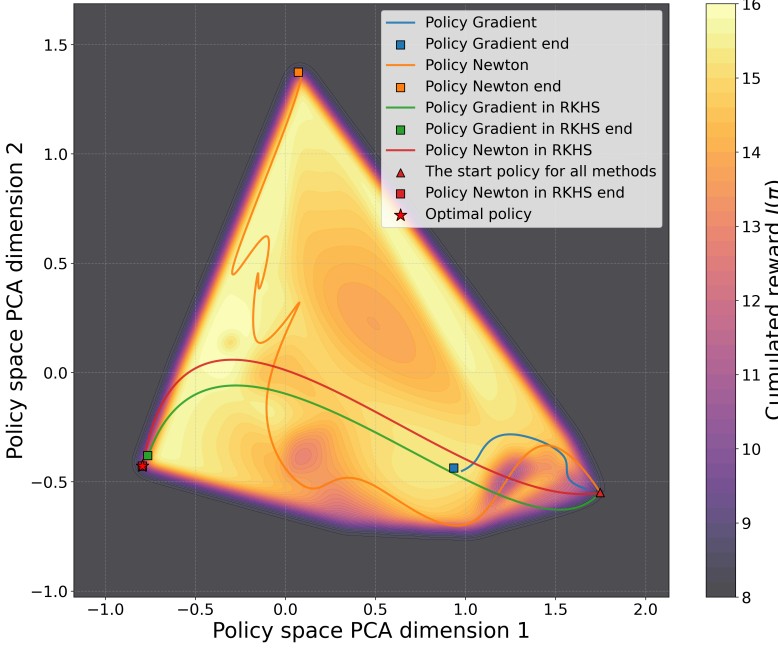

Figure 3: Alternative visualization of the policy landscape with enhanced reward contrast. The optimization trajectories are identical to Figure 1(b); only the color scale of the value surface is adjusted.

# H  OPTIMIZATION DETAILS

To solve the optimization problem formulated in Equation 7, we implemented a conjugate gradient optimization framework based on the Newton-CG method. This approach combines the second-order convergence properties of Newton's method with the computational efficiency of the conjugate gradient algorithm, making it particularly suitable for our problem where the dimensionality of the Hessian matrix $H$ scales with the volume of trajectory data $N \times T$.

Specifically, we address the following optimization problem:

$$\bar{\boldsymbol{\alpha}}^* = \underset{\bar{\boldsymbol{\alpha}} \in \mathbb{R}^{NT}}{\operatorname{argmin}} \left\{ \langle v, \bar{\boldsymbol{\alpha}} \rangle + \frac{1}{2} \langle H\bar{\boldsymbol{\alpha}}, \bar{\boldsymbol{\alpha}} \rangle + \frac{\beta}{6} \|\bar{\boldsymbol{\alpha}}\|_2^3 \right\}$$

This optimization problem incorporates a linear term $\langle v, \bar{\boldsymbol{\alpha}} \rangle$, a quadratic term $\frac{1}{2}\langle H\bar{\boldsymbol{\alpha}}, \bar{\boldsymbol{\alpha}} \rangle$, and a cubic regularization term $\frac{\beta}{6}\|\bar{\boldsymbol{\alpha}}\|_2^3$. The objective function and its gradient are computed as:

$$f(\bar{\boldsymbol{\alpha}}) = \langle v, \bar{\boldsymbol{\alpha}} \rangle + \frac{1}{2}\langle H\bar{\boldsymbol{\alpha}}, \bar{\boldsymbol{\alpha}} \rangle + \frac{\beta}{6}\|\bar{\boldsymbol{\alpha}}\|_2^3$$

$$\nabla f(\bar{\boldsymbol{\alpha}}) = v + H\bar{\boldsymbol{\alpha}} + \frac{\beta}{2}\|\bar{\boldsymbol{\alpha}}\|\bar{\boldsymbol{\alpha}}$$

In each Newton iteration, we determine the search direction by solving the linear system $(H + \frac{\beta}{2}\|\bar{\boldsymbol{\alpha}}\|\mathcal{I})\Delta\bar{\boldsymbol{\alpha}} = -\nabla f(\bar{\boldsymbol{\alpha}})$. The conjugate gradient method is employed to efficiently solve this linear system, avoiding the high computational cost of directly computing $(H + \frac{\beta}{2}\|\bar{\boldsymbol{\alpha}}\|\mathcal{I})^{-1}$. This method constructs a set of conjugate directions $\{p_i\}$ and progressively approximates the optimal solution through orthogonal projections. The algorithm proceeds as follows:

1. Initialize residual $r_0 = -\nabla f(\bar{\boldsymbol{\alpha}}_0) = -(v + H\bar{\boldsymbol{\alpha}}_0 + \frac{\beta}{2}\|\bar{\boldsymbol{\alpha}}_0\|\bar{\boldsymbol{\alpha}}_0)$ and initial search direction $p_0 = r_0$

2. For each iteration $k$:
   - Compute optimal step size $\alpha_k = \frac{r_k^T r_k}{p_k^T(H + \frac{\beta}{2}\|\bar{\boldsymbol{\alpha}}_k\|\mathcal{I})p_k}$
   - Update solution $\Delta\bar{\boldsymbol{\alpha}}_{k+1} = \Delta\bar{\boldsymbol{\alpha}}_k + \alpha_k p_k$
   - Update residual $r_{k+1} = r_k - \alpha_k(H + \frac{\beta}{2}\|\bar{\boldsymbol{\alpha}}_k\|\mathcal{I})p_k$
   - Calculate conjugate direction update coefficient $\beta_k = \frac{r_{k+1}^T r_{k+1}}{r_k^T r_k}$
   - Update search direction $p_{k+1} = r_{k+1} + \beta_k p_k$

In our implementation, we utilized the `minimize` function from the SciPy optimization library, configured with the 'Newton-CG' method. To balance optimization accuracy and computational efficiency, we set the convergence tolerance to $10^{-3}$ and the maximum number of iterations to 500.

## H.1 COMPUTATIONAL COST ANALYSIS

To address the practical concerns regarding the computational overhead of second-order methods, we present a comparison of the training time (wall-clock time) for the proposed method and baselines.

All experiments reported in this subsection were conducted on the CartPole environment. The total training duration corresponds to $1.2 \times 10^7$ training steps for each method. All runs were executed on the same hardware infrastructure (Intel Xeon Gold 5218 CPU) to ensure a fair comparison.

Table 1 summarizes the total wall-clock time required. We compare our Policy Newton in RKHS against first-order RKHS methods, as well as parametric methods with varying model complexities (Polynomial features of degree 1 and 3).

**Analysis.** The results indicate that the Policy Newton in RKHS requires approximately $1.97\times$ the computation time of its first-order counterpart (Policy Gradient in RKHS). This overhead is primarily attributed to the construction of the Hessian operator and the Conjugate Gradient (CG) iterations required to solve the Newton step.

However, it is crucial to interpret this cost in the context of sample efficiency:

Relative Overhead: The $\sim 2\times$ cost factor is consistent with the overhead observed in parametric second-order methods (e.g., Policy Newton Poly-3 vs. Gradient Poly-3), suggesting that the RKHS formulation does not introduce disproportionate computational burdens.

Thus, while computationally more intensive per update, Policy Newton in RKHS remains practically feasible for standard RL benchmarks.

Table 1: Comparison of total training time (in minutes) for $1.2 \times 10^7$ training steps on the CartPole task. The RKHS methods utilize a non-parametric representation scaling with sample size. Poly-1 and Poly-3 denote parametric policies using polynomial feature representations of degree 1 and degree 3, respectively. We note that the degree 3 is used in the experiment of the main paper.

| Method | Optimization Type | Time (min) |
|---|---|---|
| **Policy Newton in RKHS (Ours)** | Second-Order (RKHS) | 43.5 |
| Policy Gradient in RKHS | First-Order (RKHS) | 22.1 |
| Policy Newton (Poly 3) | Second-Order (Parametric) | 18.8 |
| Policy Gradient (Poly 3) | First-Order (Parametric) | 9.7 |
| Policy Newton (Poly 1) | Second-Order (Parametric) | 16.3 |
| Policy Gradient (Poly 1) | First-Order (Parametric) | 8.5 |

# I    EXTENSION TO CONTINUOUS ACTION SPACES

In the main text, we focused on discrete action spaces to establish the theoretical foundation of Policy Newton in RKHS. In this appendix, we demonstrate that our framework naturally extends to continuous action spaces. Specifically, we derive the second-order optimization steps for a Gaussian policy parameterized by a RKHS function.

## I.1    GAUSSIAN POLICY IN RKHS

We consider a continuous action space $\mathcal{A} \subseteq \mathbb{R}^d$. The policy is modeled as a multivariate Gaussian distribution with a fixed covariance matrix $\Sigma \in \mathbb{R}^{d \times d}$ and a state-dependent mean $\mu(s)$ represented by a function $h$ in a Reproducing Kernel Hilbert Space (RKHS) $\mathcal{H}_K^d$. The policy is defined as:

$$\pi_h(a \mid s) = \frac{1}{\sqrt{(2\pi)^d |\Sigma|}} \exp\left(-\frac{1}{2}(a - h(s))^\top \Sigma^{-1}(a - h(s))\right). \tag{18}$$

Here, $h = [h_1, \ldots, h_d]^\top$ where each component $h_j \in \mathcal{H}_K$ is a function in a scalar RKHS associated with kernel $K(\cdot, \cdot)$. The reproducing property implies $h(s) = \langle h, K(s, \cdot) \rangle_{\mathcal{H}_K^d}$ (component-wise).

The log-policy is given by:

$$\log \pi_h(a \mid s) = -\frac{1}{2}(a - h(s))^\top \Sigma^{-1}(a - h(s)) + C, \tag{19}$$

where $C$ is a constant independent of $h$.

## I.2    FRÉCHET DERIVATIVES IN RKHS

We derive the Fréchet derivatives of the expected return $J(\pi_h)$ with respect to the function $h$.

**First-Order Derivative.**    The gradient of the log-policy with respect to $h$ is simply the kernel function scaled by the score vector (Zhang et al., 2025):

$$\nabla_h \log \pi_h(a \mid s) = K(s, \cdot) \left[\Sigma^{-1}(a - h(s))\right] \in \mathcal{H}_K^d. \tag{20}$$

Note that this is a standard element in the RKHS. Consequently, the Policy Gradient in RKHS is:

$$\nabla_h J(\pi_h) = \mathbb{E}_{\omega \sim p(\omega; \pi_h)} \left[\sum_{t=0}^{T-1} \Psi_t(\omega) K(s_t, \cdot) \left[\Sigma^{-1}(a_t - h(s_t))\right]\right]. \tag{21}$$

**Second-Order Derivative (Hessian Operator).**    Differentiating the log-policy again yields the Hessian operator. The Gauss-Newton approximation (outer product of gradients) involves the tensor product of the gradient element with itself. The curvature term arises from the second derivative of the log-likelihood. Specifically, the Hessian of the log-policy is the operator:

$$\nabla_h^2 \log \pi_h(a \mid s) = -\left(K(s, \cdot)\Sigma^{-1}\right) \otimes K(s, \cdot). \tag{22}$$

This notation denotes an operator $T : \mathcal{H}_K^d \to \mathcal{H}_K^d$ such that for any $u \in \mathcal{H}_K^d$, $Tu = K(s, \cdot)\left[\Sigma^{-1}\langle K(s, \cdot), u\rangle_{\mathcal{H}_K^d}\right] = K(s, \cdot)[\Sigma^{-1}u(s)]$.

The full Hessian of the objective $J(\pi_h)$ is thus:

$$
\begin{aligned}
\nabla_h^2 J(\pi_h) = \mathbb{E}_{\omega \sim p(\omega; \pi_h)} &\left[ \left( \sum_{t=0}^{T-1} \Psi_t(\omega)\nabla_h \log \pi_h^t \right) \otimes \left( \sum_{t'=0}^{T-1} \nabla_h \log \pi_h^{t'} \right) \right. \\
&\left. + \sum_{t=0}^{T-1} \Psi_t(\omega) \left( -(K(s_t, \cdot)\Sigma^{-1}) \otimes K(s_t, \cdot) \right) \right].
\end{aligned}
\tag{23}
$$

### I.3 FINITE-DIMENSIONAL REDUCTION

By applying the Representer Theorem, we seek the update step in the form:

$$
\Delta h(\cdot) = \sum_{l=1}^{NT} K(x_l, \cdot)\boldsymbol{w}_l,
\tag{24}
$$

where $x_l = (s_l, a_l)$ corresponds to the $l$-th sample in the dataset, and $\boldsymbol{w}_l \in \mathbb{R}^d$ are coefficient vectors. We define the full coefficient vector $\bar{\boldsymbol{\alpha}} \in \mathbb{R}^{NTd}$ by stacking $\boldsymbol{w}_1, \dots, \boldsymbol{w}_{NT}$.

**Theorem I.1** *For a continuous Gaussian policy, the optimization of the Policy Newton step in RKHS is equivalent to minimizing:*

$$
L(\bar{\boldsymbol{\alpha}}) = \langle v, \bar{\boldsymbol{\alpha}}\rangle + \frac{1}{2}\langle H\bar{\boldsymbol{\alpha}}, \bar{\boldsymbol{\alpha}}\rangle + \frac{\beta}{6}\|\bar{\boldsymbol{\alpha}}\|_2^3.
\tag{25}
$$

*The vector $v \in \mathbb{R}^{NTd}$ is composed of blocks $v_i \in \mathbb{R}^d$ ($i = 1 \dots NT$):*

$$
v_i = \frac{1}{N}\sum_{l=1}^{NT} \Psi_l(\omega)K(s_l, s_i)\Sigma^{-1}(a_l - h(s_l)).
\tag{26}
$$

*The matrix $H \in \mathbb{R}^{NTd \times NTd}$ is given by:*

$$
H = \frac{1}{N}\sum_{l=1}^{NT} \Psi_l(\omega)\left[ G_l \otimes (\delta_l\delta_l^\top - \Sigma^{-1}) \right],
\tag{27}
$$

*where $\delta_l = \Sigma^{-1}(a_l - h(s_l)) \in \mathbb{R}^d$, and $G_l = \mathbf{k}_l\mathbf{k}_l^\top \in \mathbb{R}^{NT \times NT}$ is the outer product of the kernel column vector $\mathbf{k}_l = [K(s_l, s_1), \dots, K(s_l, s_{NT})]^\top$. The Kronecker product $A \otimes B$ here assumes the standard block layout where $A$ (the kernel matrix part) dictates the block structure and $B$ (the action dimension part) dictates the content of each block.*

## J  LLM USAGE DISCLOSURE

We used a large language model solely for writing polish. Its assistance was limited to grammar and style edits, wording suggestions for titles/abstract/captions, consistency of terminology, and minor LaTeX phrasing (e.g., figure/table captions and cross-reference text).

