# OpenReview forum: "Policy Newton Algorithm in Reproducing Kernel Hilbert Space"
_ICLR.cc/2026/Conference — ICLR 2026 Poster_

### Official Review · Reviewer_ZjiM · 2025-10-31

**Soundness:** 4
**Presentation:** 3
**Contribution:** 3
**Rating:** 6
**Confidence:** 2

**Summary:**

The paper proposes a method for utilizing second-order optimization for policies represented in Reproducing Kernel Hilbert Spaces (RKHS). Direct second-order optimization is not feasible due to the infinite-dimensional Hessian operator in RKHS. Hence, the authors introduce a finite-dimensional optimization problem, whose solution is equivalent to the Newton step. The authors compare their method to the vanilla Policy Gradient and second-order Policy Newton method, as well as the Policy Gradient in RKHS, demonstrating faster convergence (in terms of training iterations).

**Strengths:**

The authors provide extensive theory for their method, proving a quadratic convergence rate. The empirical evaluation results reflect the superior convergence rate.

**Weaknesses:**

1. The authors evaluate their method only on three tasks, which are all very low-dimensional, discrete, and relatively simple. I encourage the authors to add more tasks to the evaluation. Is the method also applicable to continuous control tasks?

2. While the proposed method achieves the highest reward out of the methods compared, it still does not seem to solve the LunarLander task consistently (the Gymnasium documentation specifies a reward threshold of 200 for an episode to be considered solved). Furthermore, all of the progress of the RKHS methods in the LunarLander task seems to happen in the first couple of iterations, which are not shown in the plot. For the rest of the training, the performance stagnates. What might be preventing the method from learning to solve the task?

3. The main drawbacks of second-order methods are the increased computation time and the limited scalability to larger models. The paper is lacking an evaluation of the computation time compared to first-order optimization. Furthermore, a comparison of the computation required for different policy sizes would be helpful.

**Questions:**

1. The introduction is relatively vague about the advantages of RKHS policy representations, simply stating that RKHS "offer a powerful non-parametric alternative, [...], valued for its representational flexibility, potential for improved sample efficiency, and capacity for dynamic adjustment during learning". Perhaps the introduction could be more explicit about what makes this representation more suitable, and in which kinds of tasks might benefit the most from these representations.

2. The description of plot 1b is too short. What exactly does the plot visualize? How is the PCA reduction done? There is no interpretation of the results. Also, the difference in reward between the optimal policy and suboptimal points is hard to assess, as large parts of the plot seem to have more or less the same color.

3. Lines 435-436 state that the "policy optimization in RKHS effectively leverages infinite-dimensional feature representations, enabling the optimization process to escape local optima". How does the feature representation help with escaping local optima?

4. What are the shaded areas in Figure 2?

5. Line 477 states that "Policy Newton in RKHS achieved significantly faster convergence to superior episodic rewards compared to first-order and parameteric Newton baselines", but some of the baselines in Figures 2(a) and (b) did not converge yet, so from the plot, it is not clear whether Policy Newton actually converges to superior episodic rewards.

Comments:

1. Specifying the training progress in "training iterations" in Figures 1 and 2 makes it hard to compare the convergence speed of the methods to other RL algorithms, consider changing the x-axis labels to environment steps.

2. Line 101 cites Maniyar et al. for the policy gradient method. The method, however, goes back to [1], which is not cited here.

[1] Ronald J. Williams "Simple statistical gradient-following algorithms for connectionist reinforcement learning." Machine learning 8.3 (1992).

---

> ### Author Response · Authors · 2025-11-22
> **Response 1**
>
> Thanks for supporting the acceptance of our work. Please find our responses to your questions below.
>
> ## Weaknesses
>
> > The authors evaluate their method only on three tasks, which are all very low-dimensional, discrete, and relatively simple. I encourage the authors to add more tasks to the evaluation. Is the method also applicable to continuous control tasks?
>
> We thank the reviewer for this constructive suggestion.
>
> **1. Applicability to Continuous Control:**
> Yes, our framework is fully applicable to continuous control. To demonstrate this rigorously, we have added a new **Appendix I** (Extension to Continuous Action Spaces). There, we derive the second-order Fréchet derivatives and the finite-dimensional reduction specifically for Gaussian policies in RKHS.
>
> **2. New Experiments on Continuous Tasks:**
> Following your encouragement, we have expanded our experimental evaluation to include two continuous control benchmarks from the Gymnasium suite: **Inverted Pendulum** and **Hopper**.
>
> The results (now integrated into Figure (c-d) and Section 5.2) show that Policy Newton in RKHS achieves superior sample efficiency and convergence speed compared to parametric baselines in these continuous settings as well. We believe these additional experiments significantly strengthen the paper's empirical claims.
>
> ---
>
> > While the proposed method achieves the highest reward out of the methods compared, it still does not seem to solve the LunarLander task consistently (the Gymnasium documentation specifies a reward threshold of 200 for an episode to be considered solved). Furthermore, all of the progress of the RKHS methods in the LunarLander task seems to happen in the first couple of iterations, which are not shown in the plot. For the rest of the training, the performance stagnates. What might be preventing the method from learning to solve the task?
>
> We appreciate the reviewer's detailed examination of the learning curves. We acknowledge that while our method outperforms the baselines, it stagnates before strictly "solving" the task (reward > 200).
>
> We attribute this behavior to two main factors related to our specific experimental design:
>
> 1.  Naive Implementation & Lack of Engineering Tricks:
>     Our primary goal was to rigorously validate the proposed theoretical framework. Therefore, we utilized a "clean" implementation without the standard engineering tricks that are essential for SOTA performance in Deep RL. Specifically, we used a simple Monte-Carlo based Critic without Generalized Advantage Estimation (GAE), value clipping, or target networks. The high variance in value estimation likely hinders the policy from fine-tuning the precise control required for a perfect landing.
>
> 2.  Premature Convergence to Local Optima:
>     The "rapid initial progress followed by stagnation" pattern noted by the reviewer is characteristic of second-order optimization methods. Policy Newton is highly efficient at exploiting gradient information to find a stationary point. However, without sophisticated exploration mechanisms (e.g., entropy annealing or intrinsic curiosity), this aggressive optimization can cause the policy to converge prematurely to a sub-optimal local maximum.
>
> In summary, our experiments successfully demonstrate the superior convergence rate predicted by our theory. Achieving the SOTA "solved" threshold would likely require combining our second-order update rule with the advanced exploration and value estimation techniques common in deep learning, which we identify as future work.
>
> ---
>
> > The main drawbacks of second-order methods are the increased computation time and the limited scalability to larger models. The paper is lacking an evaluation of the computation time compared to first-order optimization. Furthermore, a comparison of the computation required for different policy sizes would be helpful.
>
> We appreciate the suggestion to evaluate the computational cost. We conducted a wall-clock time analysis on the CartPole environment over 1.2 x 10^7 training steps and have added the detailed results to Appendix H. The data indicates that Policy Newton in RKHS (43.5 min) requires approximately twice the training time of Policy Gradient in RKHS (22.1 min), while increasing the polynomial degree of parametric baselines (Policy Newton and Policy Gradient) from 1 to 3 also resulted in a consistent increase in computation time.
>
> This overhead is expected due to the construction of the Hessian operator and the Conjugate Gradient iterations. However, we argue that this additional cost is a favorable trade-off given the significant gain in sample efficiency. In scenarios where environment interaction is expensive relative to CPU time, the ability to reach optimal rewards with fewer episodes justifies the increased wall-clock time.
>
> ---

---

> ### Author Response · Authors · 2025-11-22
> **Response 2**
>
> ## Questions
>
> > The introduction is relatively vague about the advantages of RKHS policy representations, simply stating that RKHS "offer a powerful non-parametric alternative, [...], valued for its representational flexibility, potential for improved sample efficiency, and capacity for dynamic adjustment during learning". Perhaps the introduction could be more explicit about what makes this representation more suitable, and in which kinds of tasks might benefit the most from these representations.
>
> We appreciate the reviewer's constructive comment regarding the need for a more explicit articulation of the advantages offered by RKHS policy representations.
>
> **"what makes this representation more suitable"**:
> The core suitability of RKHS policies in advanced reinforcement learning stems from two intrinsic structural properties: strong representational capability and dynamic complexity adaptation. The RKHS framework, defined in an infinite-dimensional functional space, provides universal approximation capabilities. Crucially, it facilitates dynamic complexity adaptation through the Representer Theorem, which ensures that policy updates are efficiently restricted to the finite-dimensional span of observed data points. This allows the model complexity to adapt precisely to the task requirements without pre-defined architectural constraints.
>
> **"which kinds of tasks might benefit the most"**:
> These properties are particularly advantageous in data-constrained environments, where sample efficiency is paramount, and in safety-critical applications, where the explicit decomposition of the policy into kernel evaluations centered at specific data points offers enhanced interpretability.
>
> We have revised the Introduction section of the manuscript to incorporate these clarifications.
>
> ---
>
>
> > The description of plot 1b is too short. What exactly does the plot visualize? How is the PCA reduction done? There is no interpretation of the results. Also, the difference in reward between the optimal policy and suboptimal points is hard to assess, as large parts of the plot seem to have more or less the same color.
>
> We appreciate the reviewer’s comment and have expanded the explanation of Figure 1(b) accordingly.
>
> The plot visualizes the optimization trajectories of all compared methods in the policy parameter space, after projecting the high-dimensional policy vectors onto a 2D plane using PCA. To construct this visualization, we collect all policy parameters visited during training by all methods, together with the analytically known optimal policy, and perform PCA on this combined set. The first two principal components define a shared low-dimensional subspace, and each policy iterate is projected into this space to reveal the relative motion of different algorithms toward (or away from) the optimal policy.
>
> The background color represents a smooth interpolation of the expected return evaluated at the projected sample points. Its purpose is to provide geometric context for the optimization trajectories rather than to emphasize sharp reward differences; indeed, the reward landscape around the optimum is relatively flat, which naturally limits visual contrast.
>
> In response to the reviewer’s suggestion, we have added an additional high-contrast version of this value landscape in the Appendix G, which makes the reward differences more visually distinguishable while keeping the trajectories unchanged. A detailed description of the PCA projection and interpolation procedure has also been added to the appendix for completeness.
>
> ---
>
>
> > Lines 435-436 state that the "policy optimization in RKHS effectively leverages infinite-dimensional feature representations, enabling the optimization process to escape local optima". How does the feature representation help with escaping local optima?
>
> We thank the reviewer for raising this important question. The statement refers to an optimization-side effect rather than a representational one. In the toy environment—all states and actions are finite, and all methods directly optimize the probability values for every state–action pair—all four methods have exactly the same representational capacity and are able to express the optimal policy. Therefore, the observed difference in whether an algorithm escapes suboptimal regions cannot be attributed to policy expressiveness.
>
> Instead, the key distinction lies in the optimization geometry. In an RKHS, the update step lies in the span of kernel evaluations at all visited state–action pairs, yielding a data-dependent function space whose effective dimensionality and curvature structure are richer than those of a fixed finite-dimensional parameterization. As a result, the search direction can better align with the true functional gradient and Hessian of the value landscape, allowing RKHS-based Newton updates to move out of shallow suboptimal attraction regions that trap parametric methods. We clarify this point in the revised text.
>
> ---

---

> ### Author Response · Authors · 2025-11-22
> **Response 3**
>
> > What are the shaded areas in Figure 2?
>
> We thank the reviewer for pointing this out. The shaded regions in Figure 2 correspond to the 95% confidence intervals of the episodic rewards, computed over 5 independent training runs for each method. We have added this explanation to the caption in the revised manuscript.
>
> ---
>
>
> > Line 477 states that "Policy Newton in RKHS achieved significantly faster convergence to superior episodic rewards compared to first-order and parameteric Newton baselines", but some of the baselines in Figures 2(a) and (b) did not converge yet, so from the plot, it is not clear whether Policy Newton actually converges to superior episodic rewards.
>
> We appreciate the reviewer for this nuanced observation. We agree that our original phrasing could be refined.
>
> **Clarification on Convergence vs. Performance**: In simple tasks or with extended training (as we verified with longer runs on LunarLander), parametric baselines can indeed eventually catch up to the performance of our method. In these cases, the primary advantage of Policy Newton in RKHS is its superior sample efficiency—reaching the target reward with significantly fewer interactions. We have revised Line 477 to emphasize "convergence speed" more precisely.
>
> **Why RKHS Newton Can Achieve Higher Rewards**: However, in certain complex scenarios (like the gap observed in Fig 2(a) or the Hopper task), our method does yield a strictly higher asymptotic reward. We attribute this superior performance to two key factors:
>
> (a) Curvature-Aware Optimization: First-order methods often struggle with ill-conditioned landscapes. By incorporating the Hessian operator, our method effectively rescales the update steps along different curvature directions. This allows the policy to navigate through narrow valleys and escape poor local optima that might trap standard gradient-based approaches, leading to a better final policy.
>
> (b) Non-parametric Flexibility: Unlike the baselines, which are constrained by a fixed polynomial parameterization, the RKHS policy is non-parametric. Its effective model capacity scales with the richness of the trajectory data (via the kernel). This allows it to approximate complex, non-linear optimal policy surfaces more accurately than a fixed-capacity parametric model, potentially raising the performance ceiling.
>
> ## Comments:
>
> ---
>
> > Specifying the training progress in "training iterations" in Figures 1 and 2 makes it hard to compare the convergence speed of the methods to other RL algorithms, consider changing the x-axis labels to environment steps.
>
> We have adopted this suggestion. Both Figure 1 and Figure 2 have been updated to use Environment Steps on the x-axis to facilitate direct comparison with other RL algorithms.
>
>
> ---
>
> > Line 101 cites Maniyar et al. for the policy gradient method. The method, however, goes back to [1], which is not cited here.
> > [1] Ronald J. Williams "Simple statistical gradient-following algorithms for connectionist reinforcement learning." Machine learning 8.3 (1992).
>
> We thank the reviewer for pointing out this oversight. We have updated the manuscript to include the citation for Williams (1992) to properly credit the original formulation of the Policy Gradient method.
>
>
> ---

---

### Official Review · Reviewer_6Lar · 2025-11-01

**Soundness:** 4
**Presentation:** 3
**Contribution:** 3
**Rating:** 8
**Confidence:** 3

**Summary:**

This paper studies second-order policy optimization (Newton) when the policy is modelled by a function belonging to a RKHS. It introduces a tractable method for computing (matrix-vector products with) the inverse Hessian operator, which is infinite-dimensional due to the RKHS assumption. It proves (quadratic) convergence rates for the proposed algorithm, and provides a few empirical evaluations.

**Strengths:**

The paper is written clearly, and seems to be correct.
Results are novel and interesting, the proposed method may have a strong impact.

**Weaknesses:**

I do not see any major weaknesses, however I can highlight a couple of minor issues:

- Section 4.3 seems unnecessary, it’s just a re-statement of known results about convergence rate of Newton’s method on strongly convex losses. Perhaps this space could be used instead to extend Section 3, which represents the main contribution and it’s not very easy to grasp.

- Line 94: “The objective of RL is to minimize…” It should be “maximize”. Similarly, the following equation should be “argmax”, not “argmin”

**Questions:**

- Why regularisation is cubic instead of quadratic in equation (3)?

- How does RKHS Policy Newton do in wall clock time? (Figure 1 and 2)

- How do RKHS policy methods do when state and/or actions are high-dimensional?

---

> ### Author Response · Authors · 2025-11-22
> **Response 1**
>
> Thanks for supporting the acceptance of our work. Please find our responses to your questions below.
>
> ## Weaknesses
>
> > I do not see any major weaknesses, however I can highlight a couple of minor issues:
> > > Section 4.3 seems unnecessary, it’s just a re-statement of known results about convergence rate of Newton’s method on strongly convex losses. Perhaps this space could be used instead to extend Section 3, which represents the main contribution and it’s not very easy to grasp.
>
> We thank the reviewer for this insightful suggestion. We agree that Section 3 is the core contribution of our work and necessitates a clearer exposition. We have adopted your suggestion to expand Section 3, while also clarifying the motivation for retaining Section 4.3.
>
> **1. Enhancing Section 3:**
> Following your advice, we have significantly revised Section 3 to improve readability and intuition. Specifically:
> * We added an "Intuitive Interpretation" paragraph following Theorem 3.2 (Theorem 3.1 in the original manuscript). This text helps bridge the gap between the abstract infinite-dimensional operator definitions and the practical finite-dimensional matrix construction used in the algorithm.
> * We explicitly explained the physical meaning of the terms in the Hessian matrix $H$, clarifying how the covariance of kernel sections captures the curvature information in the functional space.
> * We refined the explanation of the Representer Theorem application to better illustrate how the infinite-dimensional optimization reduces to the tractable finite-dimensional problem.
>
> **2. Rationale for Retaining Section 4.3:**
> Regarding Section 4.3, we acknowledge the reviewer's valid point that local quadratic convergence is a standard property of Newton-type methods. However, we have chosen to retain this analysis (with added context in the revised manuscript) for two specific reasons:
> * **Verification of the Infinite-to-Finite Reduction:** Our algorithm relies on transforming an optimization problem in an infinite-dimensional Hilbert space into a finite-dimensional proxy via the Representer Theorem. It is theoretically non-trivial to ensure that this specific reduction strictly preserves the local quadratic convergence rate of the original operator-theoretic problem without rigorous verification. Section 4.3 provides this necessary proof.
> * **Completeness for the RL Community:** A key goal of this work is to bridge the gap between non-parametric RKHS methods and classical second-order optimization. Explicitly proving that our RKHS-based approach achieves the same theoretical convergence speed as parametric Newton methods serves as a crucial theoretical guarantee for readers in the RL community who may be less familiar with functional analysis.
>
> We have modified the introductory text of Section 4.3 to explicitly acknowledge the parallel with classical finite-dimensional results while emphasizing the necessity of this proof within the RKHS framework.
>
> ---
>
> > > Line 94: “The objective of RL is to minimize…” It should be “maximize”. Similarly, the following equation should be “argmax”, not “argmin”
>
> Thank you for pointing this out! You are entirely correct that RL is standardly formulated as a maximization problem, and we apologize for the confusion caused by our phrasing.
>
> Our intention was to align the problem with standard optimization literature (where Newton's method is typically derived for minimization) by minimizing the negative expected reward. However, we realize we failed to state this transformation clearly in the original text, making the use of "minimize" and "argmin" appear as an error.
>
> We have revised Section 2.1 to explicitly state that we are defining the objective as a cost function (negative reward). We appreciate you catching this oversight, as this clarification significantly improves the readability of the paper.
>
> ---

---

> ### Author Response · Authors · 2025-11-22
> **Response 2**
>
> ## Questions
>
> > Why regularisation is cubic instead of quadratic in equation (3)?
>
> We thank the reviewer for this insightful question regarding our algorithmic design.
>
> The choice of cubic regularization is primarily motivated by theoretical consistency with the smoothness properties of the objective function. Specifically, our analysis relies on the assumption that the Hessian operator is Lipschitz continuous (Assumption 4.1 in the paper).
>
> According to optimization theory, when the Hessian is Lipschitz continuous, the error between the objective function and its second-order Taylor expansion is theoretically bounded by a cubic term (proportional to $\|\Delta h\|^3$, also see our proof in Appendix C.2). Therefore, adding a cubic regularization term allows the auxiliary model to serve as a rigorous global upper bound for the true objective. Minimizing this cubic-regularized surrogate ensures a guaranteed monotonic improvement in the policy, providing a solid theoretical foundation for the algorithm's convergence.
>
>
> ---
>
> > How does RKHS Policy Newton do in wall clock time? (Figure 1 and 2)
>
> We thank the reviewer for raising this important practical question. To address this, we have conducted a detailed wall-clock time analysis on the CartPole task ($1.2 \times 10^7$ steps) and added the results to Appendix H.1.
>
> The data indicates that Policy Newton in RKHS (43.5 min) requires approximately twice the training time of Policy Gradient in RKHS (22.1 min). While the second-order update incurs a computational overhead due to Hessian construction and Conjugate Gradient iterations, we argue that this is a favorable trade-off. The method's superior sample efficiency allows it to reach optimal rewards with significantly fewer environment interactions, making it particularly valuable in scenarios where data collection is more expensive than CPU time.
>
> ---
>
> >  How do RKHS policy methods do when state and/or actions are high-dimensional?
>
> We are glad you raised this point! This question actually touches upon one of the most elegant theoretical properties of our RKHS framework.
>
> We would like to highlight two positive aspects regarding high-dimensional scalability:
>
> 1.  Optimization Independence from State Dimension:
>     A key advantage of our Policy Newton algorithm is that, thanks to the Representer Theorem (Theorem 3.2), the dimensionality of the optimization problem is determined solely by the number of samples ($N \times T$), not the dimension of the state or action space. This means that computationally, our algorithm scales remarkably well: the cost of calculating the kernel matrix only grows linearly with the state dimension.
>
>
> 2.  Compatibility with Modern Representation Learning:
>     While simple kernels (like Gaussian) can be less effective in high-dimensional spaces, our optimization framework is kernel-agnostic. It is fully compatible with modern Deep Kernel learning [1][2] approaches, where a neural network maps high-dimensional states to a lower-dimensional feature space before applying the kernel. This allows our method to inherit the feature extraction power of deep learning while retaining the rigorous second-order convergence properties proven in this paper.
>
> We believe this combination—dimension-independent optimization coupled with deep kernels—is a very promising direction for future high-dimensional control tasks.
>
> [1] Wilson, A. G., Hu, Z., Salakhutdinov, R., & Xing, E. P. (2016, May). Deep kernel learning. In Artificial intelligence and statistics (pp. 370-378). PMLR.
>
> [2] Zhang, Y., Tang, H., Lin, H., & Ding, W. (2025). Residual kernel policy network: Enhancing stability and robustness in rkhs-based reinforcement learning. In The Thirteenth International Conference on Learning Representations.
>
> ---

---

### Official Review · Reviewer_Z1vx · 2025-11-04

**Soundness:** 3
**Presentation:** 3
**Contribution:** 3
**Rating:** 6
**Confidence:** 3

**Summary:**

This paper presents a policy iteration algorithm for reinforcement learning problems where the policies are formulated directly as elements of a reproducing kernel Hilbert space (RKHS). The method extends second-order optimisation algorithms to the RKHS setting by deriving computationally tractable approximations to the Hessian and the resulting optimal step direction. Theoretical guarantees are provided regarding the approximation error and convergence to an optimal solution, and experiments complement the theoretical results with demonstrations in practical settings, where the algorithm achieves superior performance in contrast to first-order methods and parametric policy iteration approaches.

**Strengths:**

* Paper is well written and follows a clear structure.
* Rigorous theoretical analysis with resulting guarantees.
* Experimental evaluations show significant performance improvements.

**Weaknesses:**

* A minimisation problem over $J(\pi_\theta)$ is introduced in Sec. 2.1. Yet, $J$ is formulated as the expected cumulative reward, which an agent should be seeking to maximise, instead of minimise. The result of the regularised Newton step in Eq. 5 also seems to be leading in a descent, instead of ascent, direction.
* Experimental evaluation is limited to a toy experiment and relatively simple classic RL problems (e.g., CartPole).
* Notation for temperature and trajectories set use the same symbol $\mathcal{T}$.
* Non-standard notation for gradient term in first expectation Eq. 4.
* The kernel for the numerical experiments is not specified. Was it a standard Gaussian or Matern kernel? The specific details should be stated in the paper, or at least in the appendix.

**Questions:**

* Was the objective $J(\pi)$ supposed to be written as the negative cumulative reward? How do you ensure the Newton step is leading in a direction that maximises the expected cumulative reward?
* Is there an alternative reference for the outer product kernel in Definition 3.1? Kubrusly and Vieira (2008) only introduce tensor products between general Hilbert spaces, not particularly RKHSs.

---

> ### Author Response · Authors · 2025-11-22
> **Response**
>
> Thanks for supporting the acceptance of our work. Please find our responses to your questions below.
>
> ## Weaknesses
>
> > A minimisation problem over $J\left(\pi_\theta\right)$ is introduced in Sec.2.1. Yet, $J$ is formulated as the expected cumulative reward, which an agent should be seeking to maximise, instead of minimise. The result of the regularised Newton step in Eq. 5 also seems to be leading in a descent, instead of ascent, direction.
>
>
> We thank the reviewer for highlighting this inconsistency. We agree that the original formulation was ambiguous by combining the standard definition of cumulative reward (maximization) with the notation of Newton-based minimization.
>
> To address this, we have revised Section 2.1 to explicitly align our problem setting with the conventions of classical optimization theory, where Newton's method is typically derived as a descent algorithm. We have clarified that we reformulate the problem as minimizing the **negative expected reward** (i.e., setting $r \leftarrow -r$).
>
> Under this unified description, the objective $J(\pi)$ represents a cost function (or loss), and the regularized Newton step correctly represents a descent direction.
>
> ---
>
> > Experimental evaluation is limited to a toy experiment and relatively simple classic RL problems (e.g., CartPole).
>
> We thank the reviewer for this constructive suggestion. We agree that demonstrating performance on more complex and diverse tasks is crucial to validate the method's effectiveness.
>
> To address this, we have significantly expanded our experimental evaluation during the revision:
>
> 1.  **New Continuous Benchmarks:** We have added experiments on Inverted Pendulum and Hopper from the Gymnasium suite. These tasks involve continuous action spaces and higher-dimensional dynamics compared to the initial CartPole/LunarLander tasks.
> 2.  **Theoretical Extension:** To support these new experiments, we provided a rigorous derivation of the second-order optimization steps for continuous Gaussian policies in RKHS (added in Appendix I).
> 3.  **Results:** The updated results (integrated into Figure 2) demonstrate that our Policy Newton in RKHS achieves superior sample efficiency and convergence speed in these challenging continuous control settings as well.
>
> ---
>
> > Notation for temperature and trajectories set use the same symbol $\mathcal{T}$.
>
> We apologize for this oversight. We have updated the manuscript to resolve this notation clash. Specifically, we have changed the symbol for the trajectory set from $\mathcal{T}$ to $\tau$ throughout the paper to clearly distinguish it from the temperature parameter.
>
> ---
>
> > Non-standard notation for gradient term in first expectation Eq. 4.
>
> Thank you for pointing this out. We have updated the notation for the gradient term in Equation 4 to $\mathcal{Z}$ to improve clarity and standardness.
>
> ---
>
> > The kernel for the numerical experiments is not specified. Was it a standard Gaussian or Matern kernel? The specific details should be stated in the paper, or at least in the appendix.
>
> We thank the reviewer for noting this omission. We clarify that we utilized a standard Gaussian kernel for all RKHS-based algorithms in our numerical experiments. We have added this specification to the beginning of Section 5 (Experimental Section) to ensure reproducibility.
>
> ---
>
> ## Questions
>
> > Was the objective $J\left(\pi\right)$ supposed to be written as the negative cumulative reward? How do you ensure the Newton step is leading in a direction that maximises the expected cumulative reward?
>
> Yes, that is correct. As clarified in the revised Section 2.1 (also see Weakness 1), we explicitly define the objective $J(\pi)$ as the expected negative cumulative reward. Under this formulation, minimizing $J(\pi)$ is mathematically equivalent to maximizing the original cumulative reward. Consequently, the Newton descent step on this cost surface inherently leads to the maximization of the expected return.
>
> ---
>
> > Is there an alternative reference for the outer product kernel in Definition 3.1? Kubrusly and Vieira (2008) only introduce tensor products between general Hilbert spaces, not particularly RKHSs.
>
> We thank the reviewer for this helpful suggestion. We agree that the previous reference was general to Hilbert spaces and lacked specificity regarding RKHS. To provide a more precise theoretical grounding, we have updated Definition 3.1 to cite the following two references, which explicitly define the tensor product structure of RKHS and their associated kernels:
>
> [1] Kumari, R., Sarkar, J., Sarkar, S., & Timotin, D. (2017). Factorizations of kernels and reproducing kernel Hilbert spaces. Integral Equations and Operator Theory.
>
> [2] Szabó, Z., & Sriperumbudur, B. K. (2018). Characteristic and universal tensor product kernels. Journal of Machine Learning Research.
>
> These references now clearly support the definition used in our derivation.
>
> ---

---

### Author Response · Authors · 2025-11-22
**General Response**

We thank the reviewers for their constructive feedback and have revised the manuscript accordingly (changes marked in blue). The major revisions are summarized below:

1. Extension to Continuous Control (Section 5.2 & Appendix I)
To address the reviewers' common concern regarding experimental scope and task complexity, we significantly expanded our evaluation to continuous control domains. We provided a rigorous theoretical derivation for continuous Gaussian policies (Appendix I) and validated the method on the **more challenging Inverted Pendulum and Hopper benchmarks**. The results demonstrate that our method achieves superior sample efficiency and convergence speed in these complex environments compared to baselines.


2. Unification of Optimization Objectives (Section 2.1)
We reformulated Section 2.1 to explicitly define the objective as minimizing the negative expected cumulative reward, ensuring mathematical consistency with the descent direction of Newton's method.


3. Computational Cost Analysis (Appendix H)
We added a wall-clock time analysis in Appendix H to transparently quantify the trade-off between the computational overhead of second-order updates and the significant gains in sample efficiency.


4. Enhanced Intuition for RKHS Optimization (Section 3)
We revised Section 3 to provide a more intuitive interpretation of the Hessian operator terms and explicitly clarified the finite-dimensional reduction process via the Representer Theorem.

5. Visualization Details (Appendix G)
We added Appendix G to provide comprehensive methodological details and high-contrast visualizations for the policy landscape analysis presented in Figure 1b.

6. Notation and Experimental Standardization
We standardized the manuscript by resolving notation conflicts (using $\tau$ for trajectories), updating x-axis labels to "Environment Steps", and specifying the use of Gaussian kernels.

---

### Meta-Review · Area_Chair_6Tgv · 2025-12-26

**Summary:**

Reviewers broadly agreed that the paper presents a technically solid and well-motivated contribution: extending second-order (Newton-type) policy optimization to policies represented in a Reproducing Kernel Hilbert Space. The theoretical development was viewed as rigorous and potentially impactful. One prominent concern was an initial inconsistency in the optimization objective, which was corrected in the revision. The experimental results were noted to be rather limited, while during the rebuttal, more results were added, which however are still on rather simple problems. The theoretical novelty is also limited, e.g., novelty beyond classical Newton-method results, strong assumptions, etc. The rebuttal addressed many of these points effectively, but the suggested decision was informed by the fact that several concerns, particularly around technical novelty,  were not fully addressed.

**Reviewer Concerns:**

1. Experiments were strengthened during the rebuttal; however, the results are still on simple tasks.
2. Technical novelty remains limited after the rebuttal

**Reviewer Scores:**

Reviewer Z1vx 6->6
Reviewer 6Lar 8->8
Reviewer ZjiM 6->6

---

### Decision · Program_Chairs · 2026-01-26

Accept (Poster)